# Robustifying Vision-Language Models via Test-Time Prompt Adaptation

**Xingyu Zhu** [1 2] **Huanshen Wu** [2] **Shuo Wang** [2 *] **Beier Zhu** [2] **Jiannan Ge** [2] **Jiaheng Zhang** [1] **Long Chen** [3]

## Abstract

Pre-trained Vision-Language Models (VLMs) such as CLIP achieve strong zero-shot generalization, but their performance degrades sharply under adversarial perturbations. Existing test-time adaptation methods typically rely on sample-level confidence heuristics, overlooking the intrinsic distributional structure of the data. This sample-centric approach limits robustness, as it fails to distinguish confident adversarial mispredictions from true semantic consistency. In this work, we observe that adversarial distortion is structurally brittle: while holistic representations are corrupted, semantic integrity is often preserved in the distribution of augmented views. Motivated by this insight, we propose RITA, a **R**obust test-t**I**me promp**T** **A**daptation framework that shifts from sample-level estimates to distribution-level alignment. Specifically, RITA employs optimal transport to align the distribution of augmented visual features with textual prototypes, mitigating adversarial outliers and rectifying cross-modal semantic misalignment. Furthermore, we introduce a dynamic cache to progressively accumulate reliable cues from the test stream for online refinement. Extensive experiments demonstrate that RITA significantly improves adversarial robustness without compromising clean accuracy.

## 1. Introduction

Vision-Language Models (VLMs) (Li et al., 2022; Alayrac et al., 2022; Li et al., 2023; Zhu et al., 2024b; 2026c) like CLIP (Radford et al., 2021), pre-trained on massive image-text pairs, have achieved remarkable zero-shot generalization. Despite this success, VLMs remain highly vulnerable to adversarial perturbations: imperceptible noise can

[1]National University of Singapore [2]University of Science and Technology of China [3]The Hong Kong University of Science and Technology. [*]Correspondence to: Shuo Wang <shuowang.edu@gmail.com>.

*Proceedings of the 43rd International Conference on Machine Learning*, Seoul, South Korea. PMLR 306, 2026. Copyright 2026 by the author(s).

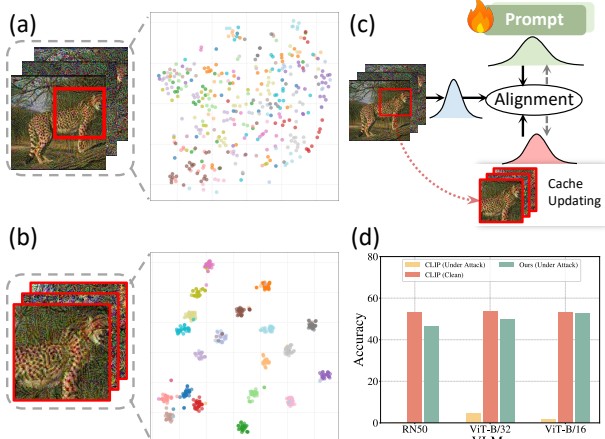

*Figure 1.* Augmented views retain more semantic cues under adversarial perturbations, enabling a cache for distribution alignment that improves adversarial performance. (a) Visualization of adversarially perturbed images, where each point represents an image and different colors denote ground-truth classes. (b) Visualization of multiple augmented views generated from the same adversarial image, colored by class label, where semantic structure partially re-emerges with improved class separability compared to (a). (c) Our method leverages the selected augmented views as a cache and aligns them with textual prompts. (d) Performance comparison across different VLM backbones, demonstrating improved robustness under adversarial attacks.

cause severe performance degradation (Szegedy et al., 2014; Madry et al., 2018; Zhu et al., 2024a; 2025b), posing security risks in real word applications.

Existing efforts to enhance the robustness of VLMs generally fall into two categories. The first line of work utilizes *adversarial training* (Mao et al., 2023; Schlarmann et al., 2024; Wang et al., 2024; Zhang et al., 2024), which aims to immunize models by explicitly integrating adversarial examples into the optimization loop. While effective, these approaches typically incur prohibitive computational costs due to on-the-fly attack generation and require access to task-specific labeled data, thereby undermining the scalability and zero-shot flexibility inherent to foundation models. The second line of work explores *test-time prompt tuning* (Yoon et al., 2024; Shu et al., 2022; Zhao et al., 2025), an efficient paradigm that adapts learnable prompt contexts or predictions during inference without modifying model parameters. However, most existing test-time methods (Wang

et al., 2025; Sheng et al., 2025) primarily rely on sample-level confidence (*e.g.*, entropy) to filter augmented views. These methods treat augmentations as isolated data points, overlooking the intrinsic distributional structure and latent semantics. Consequently, they fail to distinguish between confident adversarial mispredictions and true semantic consistency, limiting their effectiveness under attack.

To address the above limitation, we revisit how adversarial perturbations interact with test-time augmentations in VLMs. A typical white-box attack is crafted on the original input image to maximally disrupt image-text matching based on its holistic representation. This can drastically shift the attacked image embedding in the feature space and make samples from different classes highly entangle as shown in Figure 1(a). Importantly, this adversarial effect is not equally persistent across augmentations. Geometric transformations such as random cropping and flipping change the spatial correspondence of pixels, so the perturbation pattern optimized for the original configuration becomes partially mismatched after transformation. Empirically, different views exhibit heterogeneous behaviors: while some views remain strongly influenced by the attack, others still produce confident and semantically consistent predictions (Figure 1(b)). This observation suggests that robustness should be built by exploiting the distribution of augmented views and the semantic relations it contains, rather than relying solely on sample-level confidence heuristics.

Motivated by the above observation, we propose RITA, a robust test-time framework that shifts from sample-level matching to distribution-level alignment. Instead of relying on a single image embedding, RITA models the visual input as a discrete distribution over augmented views. To bridge these visual features with textual prototypes, we formulate the alignment objective as an Optimal Transport (OT) (Cuturi, 2013) problem. As illustrated in Figure 1(c), this formulation enables us to evaluate the global geometric correspondence between the visual distribution and textual representations, mitigating the influence of adversarial outliers to rectify semantic misalignment. Moreover, test samples arrive as a continuous stream, providing additional information beyond a single image. To leverage this property, RITA incorporates a dynamic cache mechanism that progressively accumulates reliable semantic views, and uses them to further refine distribution alignment online. As demonstrated in Figure 1(d), this progressive adaptation significantly enhances zero-shot robustness and maintaining competitive performance on clean data.

Extensive experiments on multiple standard benchmarks under diverse adversarial attacks demonstrate that RITA significantly enhances zero-shot robustness while preserving competitive performance on clean data. Our contributions are summarized as follows:

- We propose RITA, a robust test-time prompt adaptation framework that leverages **augmented views** to rectify adversarial misalignment at the distribution level.

- We formulate cross-modal semantic alignment as an Optimal Transport problem, complemented by a dynamic cache for progressive refinement.

- We conduct comprehensive evaluations, showing that RITA consistently outperforms existing test-time adaptation methods.

## 2. Related Work

**Adversarial Defense in VLMs.** The vulnerability of VLMs (*e.g.*, CLIP (Radford et al., 2021)) to adversarial perturbations remains a critical challenge (Dong et al., 2018; Madry et al., 2018; Zhao et al., 2023; Zhu et al., 2026d). Attacks have evolved from uni-modal perturbations (Carlini & Wagner, 2017a; Xie et al., 2019) to multi-modal strategies like Co-Attack (Zhang et al., 2022a) that disrupt cross-modal alignment. To enhance robustness, prior defenses utilize training-time strategies, primarily adversarial contrastive tuning (Mao et al., 2023; Schlarmann et al., 2024; Wang et al., 2024). However, these methods typically require expensive re-training and labeled data, limiting practicality. Consequently, inference-time robustness has emerged to secure models without weight updates, such as diffusion purification (Feng et al., 2023; 2025) and optimization-based methods (Wang et al., 2021; Zhang et al., 2022b). Notably, Test-Time Prompt Tuning approaches like TAPT (Wang et al., 2025) and R-TPT (Sheng et al., 2025) adapt prompts using unlabeled data via contrastive learning or entropy minimization. Nevertheless, these paradigms typically treat visual inputs as isolated points, failing to exploit the underlying distributional geometry of adversarial samples. In contrast, we propose RITA, a test-time prompt tuning framework that shifts from point-level alignment to distribution-level modeling.

**Optimal transport.** Optimal Transport (OT) (Cuturi, 2013) provides a principled way to compare probability distributions by accounting for the geometry of the underlying feature space. With efficient solvers such as Sinkhorn (Altschuler et al., 2017; Mensch & Peyré, 2020), OT has been widely used in generative modeling (Arjovsky et al., 2017), structural alignment (Xu et al., 2019), and domain adaptation (Courty et al., 2016). In vision–language learning, OT has also been applied to reduce semantic misalignment, including few-shot learning (Lazarou et al., 2021), distribution calibration (Guo et al., 2022; Damodaran et al., 2018; Zhu et al., 2025a;b), and prompt learning (Chen et al., 2023; Wang et al., 2023; Ren et al., 2025). For example, PLOT (Chen et al., 2023) aligns image features with multiple prompts via OT-based matching to capture

diverse semantics, while ALIGN (Wang et al., 2023) further introduces hierarchical/token-level transportation for fine-grained cross-modal alignment (Zhu et al., 2026a;b). AWT (Zhu et al., 2024c) similarly formulates image-text distance as an OT problem to model semantic correlations in the joint space. However, these approaches primarily focus on representation enhancement in clean settings. They rely on undistorted visual manifolds and do not account for the severe structural perturbations caused by adversarial attacks. Differing from these approaches, RITA repurposes OT for adversarial defense, aiming to reconstruct the distribution-level correspondence disrupted by attacks. This enables inference-time correction of structural misalignment without parameter updates, ensuring robust deployment.

## 3. Method

An overview of RITA is illustrated in Figure 2. We first introduce the preliminaries in Sec. 3.1, then detail our distributed feature modeling and dynamic distribution alignment in Sec. 3.2 and 3.3, respectively. Finally, we provide a theoretical justification in Sec. 3.4.

### 3.1. Preliminary

**Test-time prompt tuning.** Test-Time Prompt Tuning (TPT) (Shu et al., 2022) improves the zero-shot generalization of CLIP (Radford et al., 2021) by adapting textual prompts at inference time, without accessing labeled data or updating CLIP parameters. Given a test image $x_t$, TPT constructs a set of $N$ stochastic augmentations $\mathcal{X}_t = \{x_t^n\}_{n=1}^N$. For each augmented view $x_t^n$, the visual representation is $\mathbf{x}_t^n = \Phi_{\text{img}}(x_t^n)$, where $\Phi_{\text{img}}(\cdot)$ denotes the CLIP image encoder. On the text side, the prompt for class $k$ is formulated as $z_k = \{\omega_1, \omega_2, \ldots, \omega_L, c_k\}$, where $c_k$ is the token embedding of the class name, and $\omega = \{\omega_\ell\}_{\ell=1}^L$ are learnable context vectors shared across classes. The textual representation is $\mathbf{z}_k = \Phi_{\text{text}}(z_k)$, with $\Phi_{\text{text}}(\cdot)$ being the CLIP text encoder. For each view $x_t^n$, the prediction probability is computed by feature matching:

$$p(k \mid x_t^n; \omega) = \frac{\exp\big(\cos(\mathbf{x}_t^n, \mathbf{z}_k)/\tau\big)}{\sum_{j=1}^K \exp\big(\cos(\mathbf{x}_t^n, \mathbf{z}_j)/\tau\big)}, \quad (1)$$

where $\tau$ is the temperature parameter and $K$ is the number of categories. Specifically, TPT employs a confidence selection strategy to filter out unreliable augmentations. It selects a subset $\mathcal{S}$ of low-entropy views to compute the aggregated prediction $\bar{p}(k \mid x_t; \omega) = \frac{1}{|\mathcal{S}|} \sum_{n \in \mathcal{S}} p(k \mid x_t^n; \omega)$. The prompt $\omega$ is then optimized by minimizing the entropy of this aggregated distribution $\bar{p}$:

$$\omega^* = \operatorname*{argmin}_\omega \Big(-\sum_{k=1}^K \bar{p}(k \mid x_t; \omega) \log \bar{p}(k \mid x_t; \omega)\Big). \quad (2)$$

**Optimal transport.** Optimal Transport (OT) (Cuturi, 2013) provides a principled way to measure the discrepancy between two probability distributions. Consider two discrete distributions in the feature space, $\mathbb{P} = \sum_{n=1}^N a^n \delta_{\mathbf{x}^n}$ and $\mathbb{Q} = \sum_{m=1}^M b^m \delta_{\mathbf{z}^m}$, where $\delta_{\mathbf{v}}$ denotes the Dirac delta function at location $\mathbf{v}$, and $\mathbf{a} \in \Delta_N, \mathbf{b} \in \Delta_M$ are probability vectors. Given a cost matrix $\mathbf{C} \in \mathbb{R}^{N \times M}$, where $\mathbf{C}_{nm}$ measures the transport cost from $\mathbf{x}^n$ to $\mathbf{z}^m$, the entropy-regularized OT distance is defined as:

$$d_{\text{OT}}(\mathbb{P}, \mathbb{Q}; \mathbf{C}) = \min_{\mathbf{T} \in \Pi(\mathbf{a}, \mathbf{b})} \langle \mathbf{T}, \mathbf{C} \rangle - \lambda h(\mathbf{T}), \quad (3)$$

where $\Pi(\mathbf{a}, \mathbf{b}) = \{\mathbf{T} \in \mathbb{R}_+^{N \times M} \mid \mathbf{T} \mathbb{1}_M = \mathbf{a}, \mathbf{T}^\top \mathbb{1}_N = \mathbf{b}\}$ is the transport polytope, $h(\mathbf{T}) = -\sum_{n,m} T_{nm} \log T_{nm}$ is the entropic regularization term, and $\lambda \geq 0$ controls the regularization strength. This formulation enables efficient computation via the Sinkhorn algorithm.

### 3.2. Distributed Features Modeling

In this section, we present our core strategy for robust test-time inference. Moving beyond vulnerable holistic image embeddings, we model the adversarial image and textual prompts as discrete distributions and align them structurally using optimal transport.

**Adversarial perturbations.** Adversarial attacks aim to degrade model predictions by introducing small, carefully crafted perturbations. In a white-box setting, given a clean image $x$ and its ground-truth label $y$, the adversarial example $x'$ is generated by maximizing the cross-entropy loss within an $\ell_p$-norm constraint:

$$x' = \operatorname*{argmax}_{\|x'-x\|_p \leq \epsilon_{\text{adv}}} \mathcal{L}_{\text{CE}}\Big(p(\cdot \mid x'; z), y\Big), \quad (4)$$

where $\epsilon_{\text{adv}}$ denotes the perturbation budget, and $p(\cdot \mid x'; z)$ represents the probability distribution over $K$ classes as defined in Eq. (1). Typically, we approximate this optimization using the iterative Projected Gradient Descent (PGD) (Madry et al., 2018).

**Multi-prototype distribution alignment.** While adversarial attacks distort alignment at the global representation level, relying on single-point embeddings is insufficient to capture the semantic variations. To alleviate this, we construct sets of diverse visual and textual representations and model their correspondence at the distribution level.

Specifically, given an adversarial image $\hat{x}_t$, we apply data augmentations to obtain $N$ views $\{\hat{x}_t^n\}_{n=1}^N$, producing visual features $\{\hat{\mathbf{x}}_t^n\}_{n=1}^N$. Instead of a single text prototype, for each class $k$, we construct $M$ prompts $\{z_k^{(m)}\}_{m=1}^M$ using learnable context vectors, yielding textual features $\{\mathbf{z}_k^m\}_{m=1}^M$. We model the adversarial image and the $k$-th

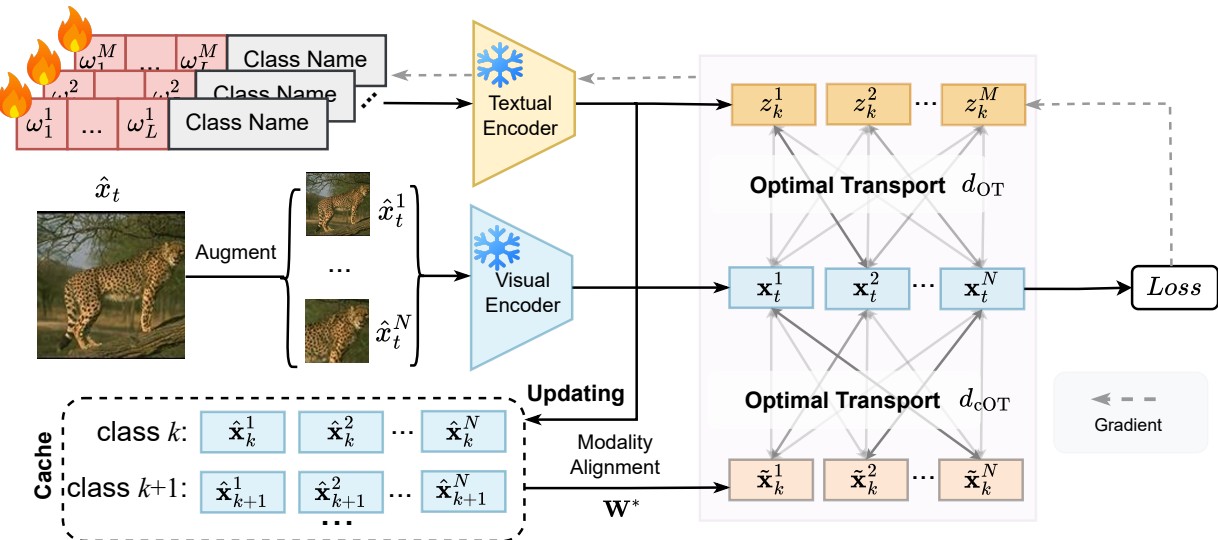

*Figure 2.* Overview of the proposed RITA framework. Given an adversarial test image, RITA extracts multi-view visual features and class-specific textual prototypes using a frozen CLIP encoder. Both modalities are modeled as discrete distributions and aligned via entropy-regularized optimal transport. Low-entropy views are used to update a dynamic cache of reliable semantics.

class prototype as discrete distributions:

$$\mathbb{P}_t = \sum_{n=1}^{N} \frac{1}{N} \delta_{\hat{\mathbf{x}}_t^n}, \quad \mathbb{Q}_k = \sum_{m=1}^{M} \frac{1}{M} \delta_{\mathbf{z}_k^m}. \quad (5)$$

Here, we assign uniform weights to both modalities, *i.e.*, $\mathbf{a}_t = \frac{1}{N}\mathbb{1}_N$ and $\mathbf{b}_k = \frac{1}{M}\mathbb{1}_M$, such that the marginal constraints of OT are satisfied. We then measure the distribution-level alignment between the visual distribution $\mathbb{P}_t$ and the textual prototype distribution $\mathbb{Q}_k$ via entropy-regularized optimal transport:

$$d_{\mathrm{OT}}(\mathbb{P}_t, \mathbb{Q}_k; \mathbf{C}_{t,k}) = \min_{\mathbf{T}_{t,k} \in \Pi} \left( \langle \mathbf{T}_{t,k}, \mathbf{C}_{t,k} \rangle - \lambda h(\mathbf{T}_{t,k}) \right), \quad (6)$$

where $\mathbf{T}_{t,k} \in \mathbb{R}_+^{N \times M}$ is the transport plan, and the cost matrix is defined by the cosine distance, $\mathbf{C}_{t,k}(n, m) = 1 - \cos(\hat{\mathbf{x}}_t^n, \mathbf{z}_k^m)$. Intuitively, a smaller OT distance indicates better alignment between the test image and class $k$ at the distribution level. Accordingly, we predict the label by selecting the class with the minimum transport cost:

$$\hat{y} = \operatorname*{argmin}_{k \in [K]} d_{\mathrm{OT}}(\mathbb{P}_t, \mathbb{Q}_k; \mathbf{C}_{t,k}). \quad (7)$$

### 3.3. Dynamic Distribution Alignment

While distribution-level alignment in Sec. 3.2 effectively processes individual samples, it treats inference steps in isolation, neglecting the semantic consensus in the continuous test stream. To exploit this temporal information, we introduce a dynamic cache mechanism that accumulates reliable visual features to iteratively refine the alignment online.

**Confidence-based cache update.** To update the cache with reliable samples, we evaluate the confidence of each individual augmented view. Instead of solving the full global transport problem, we quantify the instance-level alignment by measuring the average affinity between the view $\hat{\mathbf{x}}_t^n$ and the text distribution of class $k$:

$$p_t^n(k) = \frac{\exp\left( \frac{1}{M} \sum_{m=1}^{\tau M} \cos(\hat{\mathbf{x}}_t^n, \mathbf{z}_k^m) \right)}{\sum_{j=1}^{K} \exp\left( \frac{1}{M} \sum_{m=1}^{\tau M} \cos(\hat{\mathbf{x}}_t^n, \mathbf{z}_j^m) \right)}. \quad (8)$$

We compute entropy $H(p_t^n) = -\sum_{k=1}^{K} p_t^n(k) \log p_t^n(k)$ and retain confident views:

$$\mathcal{B}_t = \left\{ (\hat{\mathbf{x}}_t^n, \hat{y}_t^n) \ \middle| \ H(p_t^n) \le \gamma, \ \hat{y}_t^n = \operatorname*{argmax}_k p_t^n(k) \right\}. \quad (9)$$

We maintain a class-wise cache $\{\hat{\mathbf{X}}_k\}_{k=1}^{K}$, where $\hat{\mathbf{X}}_k \in \mathbb{R}^{N_k \times d}$ stores cached visual features pseudo-labeled as class $k$ ($N_k$ is the current cache size). The cache is updated online by prioritizing lower-entropy samples. To bridge the modality gap, we align cached visual features to the textual space of class $k$ by solving an Orthogonal Procrustes (Ouali et al., 2023) problem:

$$\mathbf{W}_k^* = \operatorname*{argmin}_{\mathbf{W}^\top \mathbf{W} = \mathbf{I}} \left\| \hat{\mathbf{X}}_k \mathbf{W} - \mathbf{1}_{N_k} \bar{\mathbf{z}}_k^\top \right\|_F^2, \quad (10)$$

where $\bar{\mathbf{z}}_k = \frac{1}{M} \sum_{m=1}^{M} \mathbf{z}_k^m$ is the mean text embedding for class $k$. The aligned cached features are $\tilde{\mathbf{X}}_k = \hat{\mathbf{X}}_k \mathbf{W}_k^*$.

**Cache-based distribution matching.** For each class $k$, we instantiate a discrete probability distribution over the

aligned cached features:

$$\tilde{\mathbb{Q}}_k = \frac{1}{N_k} \sum_{j=1}^{N_k} \delta_{\tilde{\mathbf{x}}_k^j}, \qquad \text{where } \tilde{\mathbf{x}}_k^j \in \tilde{\mathbf{X}}_k. \quad (11)$$

We quantify the discrepancy via the same entropy-regularized OT distance as in Sec. 3.2, but with a cache-specific cost matrix $\tilde{\mathbf{C}}_{t,k}(n,j) = 1 - \cos(\hat{\mathbf{x}}_t^n, \tilde{\mathbf{x}}_k^j)$, and denote it as $d_{\text{cOT}}(\mathbb{P}_t, \tilde{\mathbb{Q}}_k; \tilde{\mathbf{C}}_{t,k})$.

**Final inference.** We classify the test sample by identifying the category that minimizes the joint transport cost, which integrates both the global prompt alignment and the local cache consensus:

$$\hat{y} = \underset{k \in [K]}{\arg\min} \left( d_{\text{OT}}(\mathbb{P}_t, \mathbb{Q}_k; \mathbf{C}_{t,k}) + \alpha \, d_{\text{cOT}}(\mathbb{P}_t, \tilde{\mathbb{Q}}_k; \tilde{\mathbf{C}}_{t,k}) \right),$$
$$(12)$$

where $\alpha \geq 0$ controls the contribution of the dynamic cache.

### 3.4. Theoretical Analysis

Standard methods like TPT (Shu et al., 2022) and R-TPT (Sheng et al., 2025) optimize centroid alignment via mean pooling and cosine similarity. For $\ell_2$-normalized CLIP features, this is equivalent to minimizing the squared Euclidean distance between centroids ($\|\mathbf{x} - \mathbf{z}\|^2 = 2(1 - \cos(\mathbf{x}, \mathbf{z}))$). To reveal RITA's geometric advantage, we analyze alignment using the 2-Wasserstein distance ($W_2$). Let $\mathbb{P}_t (\boldsymbol{\mu}_x, \boldsymbol{\Sigma}_x)$ and $\mathbb{Q}_k (\boldsymbol{\mu}_z, \boldsymbol{\Sigma}_z)$ be the visual and textual distributions. We obtain the following decomposition:

**Theorem 3.1** (Decomposition of Alignment Objective). *The Optimal Transport objective ($\mathcal{L}_{\text{OT}}$) imposes a stricter bound by decomposing into a centroid alignment term and a structural variance penalty:*

$$\underbrace{W_2^2(\mathbb{P}_t, \mathbb{Q}_k)}_{\mathcal{L}_{\text{OT}} \text{ (RITA)}} \approx \underbrace{\|\boldsymbol{\mu}_x - \boldsymbol{\mu}_z\|^2}_{\mathcal{L}_{\text{mean}} \text{ (TPT-equivalent)}} + \underbrace{\mathfrak{B}^2(\boldsymbol{\Sigma}_x, \boldsymbol{\Sigma}_z)}_{\mathcal{R}_{\text{var}} \text{ (Structural Penalty)}},$$
$$(13)$$

*where $\mathfrak{B}^2(\mathbf{A}, \mathbf{B}) = \text{Tr}(\mathbf{A} + \mathbf{B} - 2(\mathbf{A}^{1/2}\mathbf{B}\mathbf{A}^{1/2})^{1/2})$ is the Bures metric, quantifying the geometric mismatch between covariances.*

Equation (13) holds exactly for Gaussian distributions and serves as a general lower bound, as derived in Appendix A. This inequality highlights a critical robustness gap: minimizing only $\mathcal{L}_{\text{mean}}$ leaves structural variance $\mathcal{R}_{\text{var}}$ unconstrained, allowing attackers to distort distribution geometry.

## 4. Experiments

### 4.1. Setup

**Datasets.** We evaluate our method on eight image classification benchmarks spanning a wide range of visual domains, including generic object recognition (Caltech101 (Fei-Fei

et al., 2004)), texture recognition (DTD (Cimpoi et al., 2014)), satellite imagery (EuroSAT (Helber et al., 2019)), human action recognition (UCF101 (Soomro et al., 2012)), as well as several fine-grained classification tasks, namely Pets (Parkhi et al., 2012), Cars (Krause et al., 2013), Flowers (Nilsback & Zisserman, 2008), and Aircraft (Maji et al., 2013). To further assess robustness under distribution shifts, we conduct additional evaluations on ImageNet (Deng et al., 2009) and four of its variants that share the same label space: ImageNetV2 (Recht et al., 2019), ImageNet-Sketch (Wang et al., 2019), ImageNet-A (Hendrycks et al., 2021b), and ImageNet-R (Hendrycks et al., 2021a). These benchmarks introduce significant variations in image sources, styles, and underlying visual statistics. Detailed analysis of these datasets is provided in Appendix C.1.

**Implementation details.** We build all experiments on the official pre-trained CLIP models with two backbones, CLIP-ViT-B/32 and CLIP-ViT-B/16. We generate adversarial examples using PGD (Madry et al., 2018) under an $L_\infty$ constraint. We use $\epsilon = 4.0$ with 7 steps for both backbones. At test time, we update only the prompt parameters while keeping the CLIP backbone frozen. The prompt number $M$ is set to 4 and initialized with the template *"a photo of a"*. We use AdamW and fix the Test-Time Adaptation (TTA) step at 1 per test sample, with a learning rate of 0.005. We apply standard test-time augmentations for images, including random cropping, resizing, and horizontal flipping. For text, we use an Large Language Model (LLM) to generate class-specific descriptions(Zhu et al., 2024c). For each test image, we sample $N = 64$ augmented views (including the original). For cache construction, We set the entropy threshold to $\gamma = 0.8$. More experiment details are provided in Appendix B.

**Comparison methods.** We compare RITA with CLIP-based test-time adaptation baselines, including TPT (Shu et al., 2022), R-TPT (Sheng et al., 2025), C-TPT (Yoon et al., 2024), and MTA (Zanella & Ayed, 2024), as well as the zero-shot CLIP baseline. We also report an **Ensemble** baseline that averages predictions over augmented views. To further evaluate compatibility with robust pre-trained weights, We incorporate three representative adversarially fine-tuned CLIP models, namely TeCoA (Mao et al., 2023), PMG (Wang et al., 2024), and FARE (Schlarmann et al., 2024). All methods follow the instance-level test-time adaptation protocol: each test sample is adapted and predicted independently, without access to other test samples.

### 4.2. Main Results

**Results on fine-grained datasets.** We evaluate our method on eight fine-grained benchmark datasets with ViT-B/32 and ViT-B/16 backbones, as presented in Table 1. The results show that RITA achieves the highest adversarial accuracy

*Table 1.* Results (%) of adaptation methods on fine-grained classification datasets with $\epsilon$ set to 1.0. **Bold** and underlined entries indicate the best and second-best results, respectively. Acc. denotes accuracy on clean data, and Rob. denotes accuracy under adversarial perturbations.

| | Method | Caltech101 | | Pets | | Cars | | Flower102 | | Aircraft | | DTD | | EuroSAT | | UCF101 | | Avg. | |
|---|---|---|---|---|---|---|---|---|---|---|---|---|---|---|---|---|---|---|---|
| | | Acc. | Rob. | Acc. | Rob. | Acc. | Rob. | Acc. | Rob. | Acc. | Rob. | Acc. | Rob. | Acc. | Rob. | Acc. | Rob. | Acc. | Rob. |
| ViT-B/32 | CLIP | 90.9 | 25.6 | 83.0 | 2.1 | 49.7 | 0.0 | 65.8 | 2.2 | 18.3 | 0.0 | 40.8 | 5.4 | 18.6 | 0.1 | 62.1 | 2.4 | 53.6 | 4.7 |
| | Ensemble | 91.6 | 85.4 | 85.0 | 73.4 | 57.8 | 39.6 | 67.4 | 57.5 | 20.1 | 14.4 | 46.1 | 39.3 | 32.5 | 23.6 | 61.6 | 53.5 | 57.8 | 48.3 |
| | TPT | 91.4 | 77.6 | 84.1 | 58.6 | 62.9 | 30.8 | 63.8 | 43.6 | 19.0 | 10.1 | 42.2 | 28.6 | **35.1** | 15.1 | 62.3 | 38.1 | 57.6 | 37.8 |
| | C-TPT | 91.8 | 75.9 | 84.9 | 52.2 | 60.8 | 27.1 | 65.9 | 42.1 | 17.7 | 8.7 | 44.3 | 27.1 | 34.7 | 9.0 | 62.6 | 35.3 | 57.8 | 34.6 |
| | MTA | 91.8 | 80.8 | 85.8 | 62.6 | **64.1** | 34.5 | 64.8 | 44.8 | **20.4** | 11.1 | 44.0 | 29.3 | 34.5 | 7.8 | **63.6** | 40.1 | **58.6** | 38.8 |
| | R-TPT | 90.6 | 86.2 | 84.5 | 73.1 | 63.1 | **44.6** | 62.6 | 53.1 | 19.1 | 12.9 | 42.1 | 36.7 | 32.0 | 22.4 | 62.8 | 54.2 | 57.1 | 47.9 |
| | RITA | **92.3** | **86.5** | **85.9** | **74.5** | 59.6 | 42.9 | **68.7** | **58.7** | 20.2 | **15.2** | **46.2** | **40.1** | 33.4 | **24.8** | 62.8 | **55.0** | **58.6** | **49.7** |
| ViT-B/16 | CLIP | 85.9 | 10.8 | 83.5 | 0.5 | 55.7 | 0.0 | 61.7 | 0.1 | 15.7 | 0.0 | 40.4 | 2.4 | 23.7 | 0.0 | 58.9 | 0.5 | 53.2 | 1.8 |
| | Ensemble | 92.1 | 87.4 | 88.7 | 77.2 | 63.2 | 46.7 | 70.8 | 59.9 | 25.9 | 17.9 | 50.9 | 43.2 | 32.9 | 26.7 | 64.6 | 54.3 | 61.1 | 51.6 |
| | TPT | 94.1 | 79.6 | 87.4 | 62.8 | 66.5 | 35.5 | 66.1 | 48.3 | 23.4 | 12.3 | 45.9 | 29.1 | **42.6** | 7.4 | **67.9** | 39.7 | 61.7 | 39.3 |
| | C-TPT | 93.9 | 76.5 | 88.2 | 55.8 | 65.8 | 30.5 | 69.6 | 45.5 | 23.9 | 9.8 | 45.9 | 26.6 | 42.3 | 7.1 | 65.6 | 34.7 | 61.9 | 35.8 |
| | MTA | **94.3** | 81.9 | 88.0 | 64.5 | **67.7** | 38.2 | 65.0 | 46.9 | 24.0 | 12.6 | 46.5 | 28.7 | 42.5 | 13.7 | 67.5 | 40.8 | 61.9 | 40.9 |
| | R-TPT | 93.7 | 87.8 | 87.2 | 74.7 | 67.0 | 46.9 | 68.7 | 55.7 | 23.9 | 17.3 | 46.4 | 39.7 | 34.7 | 26.8 | 67.2 | 55.4 | 61.1 | 50.5 |
| | RITA | 93.8 | **88.5** | **89.8** | **77.3** | 64.2 | **47.1** | **71.6** | **61.3** | **26.2** | **19.2** | **51.5** | **44.7** | 33.4 | **27.6** | 65.5 | **55.8** | **62.0** | **52.7** |

*Table 2.* Classification accuracy (%) on 8 datasets using different adversarially finetuned CLIP models.

| Method | Caltech101 | | Pets | | Cars | | Flower102 | | Aircraft | | DTD | | EuroSAT | | UCF101 | | Avg. | |
|---|---|---|---|---|---|---|---|---|---|---|---|---|---|---|---|---|---|---|
| | Acc. | Rob. | Acc. | Rob. | Acc. | Rob. | Acc. | Rob. | Acc. | Rob. | Acc. | Rob. | Acc. | Rob. | Acc. | Rob. | Acc. | Rob. |
| TeCoA | 77.6 | 64.3 | 59.8 | 39.9 | 20.6 | 9.3 | 37.1 | 23.9 | 5.7 | 2.8 | 24.1 | 14.4 | **15.9** | **12.4** | 40.5 | 23.4 | 35.2 | 23.8 |
| + Ensemble | 76.0 | 68.3 | 61.3 | 54.8 | 19.4 | 14.5 | 38.4 | 33.2 | 6.9 | 4.9 | 27.5 | 25.2 | 11.5 | 12.0 | 39.4 | 34.5 | 35.1 | 30.9 |
| + MTA | **79.5** | 56.5 | 61.6 | 28.8 | 20.9 | 10.4 | 37.2 | 24.0 | 6.6 | 3.8 | 25.8 | 19.7 | 12.2 | 11.2 | **42.0** | 17.3 | 35.7 | 21.4 |
| + R-TPT | 77.1 | 70.3 | 60.9 | 54.0 | **23.1** | **17.8** | 35.4 | 30.8 | 7.0 | 4.7 | 26.7 | 24.7 | 12.0 | 11.8 | 40.1 | 35.0 | 35.3 | 31.1 |
| + RITA | 78.1 | **70.4** | **62.3** | **54.9** | 21.1 | 15.7 | **39.2** | **34.0** | **7.1** | **5.5** | **27.8** | **25.5** | 12.2 | 12.1 | 40.9 | **35.8** | **36.1** | **31.7** |
| PMG | **82.3** | 70.8 | 61.8 | 41.4 | **24.7** | 12.5 | 36.2 | 25.3 | 5.2 | 2.9 | 22.8 | 16.0 | **17.1** | 12.8 | 42.6 | 27.8 | 36.5 | 26.1 |
| + Ensemble | 78.9 | 71.7 | 60.7 | 54.1 | 16.6 | 11.6 | 37.8 | 32.7 | 7.2 | 5.1 | 26.6 | 25.2 | 14.0 | 13.9 | 42.0 | 37.7 | 35.5 | 31.5 |
| + MTA | 79.5 | 65.4 | 61.8 | 31.3 | 17.9 | 12.9 | 36.7 | 21.6 | 5.8 | 3.3 | 22.9 | 18.7 | 13.8 | 12.5 | 43.1 | 22.6 | 35.1 | 23.5 |
| + R-TPT | 79.3 | 73.2 | 62.0 | 55.1 | 18.3 | **14.9** | 35.3 | 30.4 | 5.4 | 4.1 | 25.4 | 23.0 | 13.2 | 12.9 | 42.3 | 38.2 | 35.2 | 31.4 |
| + RITA | 79.9 | **73.7** | **62.5** | **55.4** | 18.9 | 13.9 | **38.7** | **33.9** | **7.5** | **5.4** | **27.4** | **25.4** | 14.4 | **14.1** | **43.5** | **38.6** | **36.6** | **32.6** |
| FARE | 86.6 | 62.9 | 77.7 | 38.1 | 40.4 | 9.7 | 48.7 | 22.5 | 10.2 | 2.3 | 32.4 | 18.1 | **22.4** | 11.0 | 52.9 | 22.2 | 46.4 | 23.3 |
| + Ensemble | 86.1 | 80.2 | 77.9 | 70.1 | 38.8 | 29.5 | 48.9 | 42.3 | 10.5 | 7.8 | 36.9 | 32.4 | 13.5 | 11.6 | 52.8 | 45.6 | 45.6 | 39.9 |
| + MTA | **87.7** | 70.0 | 78.4 | 45.0 | 40.6 | 24.8 | 49.2 | 30.5 | 11.0 | 6.6 | 32.8 | 25.3 | 13.5 | 11.8 | 53.9 | 29.8 | 45.8 | 30.4 |
| + R-TPT | 86.5 | 81.1 | 77.4 | 70.1 | **43.0** | **33.0** | 46.2 | 40.2 | 10.0 | 7.4 | 33.6 | 30.0 | 12.9 | 12.1 | 53.8 | 46.8 | 45.4 | 40.0 |
| + RITA | 86.8 | **81.9** | **78.7** | **70.5** | 41.2 | 31.2 | **50.1** | **43.1** | **11.8** | **8.6** | **37.5** | **32.9** | 14.2 | **12.7** | 54.3 | 47.0 | **46.8** | **41.0** |

on nearly all datasets. Compared to vanilla CLIP, RITA improves average robustness by 45.0% and 50.9% with ViT-B/32 and ViT-B/16, respectively. This effectively mitigates the severe vulnerability of the baseline under adversarial attacks, which can cause near-zero accuracy on datasets such as Cars and Aircraft. Furthermore, compared to R-TPT, the state-of-the-art test-time adaptation method, RITA achieves consistent gains in average robust accuracy, outperforming it by 1.8% and 2.2% on the two architectures. Meanwhile, our method also achieves the highest average accuracy on clean samples. This demonstrates that RITA enhances adversarial robustness while preserving recognition performance in attack-free environments with negligible degradation. Notably, RITA performs better with ViT-B/16 than with ViT-B/32, aligning with the intuition that a fine-grained ViT backbone yields stronger recognition capabilities.

**Results with adversarially finetuned CLIP models.** As an inherently plug-and-play framework, we integrated RITA with three representative adversarially fine-tuned models, with results summarized in Table 2. RITA exhibits significant synergy with these robust baselines, substantially enhancing robust accuracy across all datasets without sacrificing clean performance. Notably, when combined with FARE, RITA achieves a striking 81.9% robust accuracy on Caltech101, representing a 19.0% improvement over the baseline. These results demonstrate that RITA is a versatile adaptation framework that can seamlessly fortify existing adversarial defense models during inference.

**Results under different attack types.** Table 3 further assesses the generalizability of RITA against CW (Carlini & Wagner, 2017b) and DI (Xie et al., 2019) attacks. RITA consistently demonstrates superior defense across protocols,

*Table 3.* Results (%) of adaptation methods on fine-grained classification datasets under different attacks using ViT-B/16 with $\epsilon = 1.0$.

| | Method | Caltech101 Acc. | Rob. | Pets Acc. | Rob. | Cars Acc. | Rob. | Flower102 Acc. | Rob. | Aircraft Acc. | Rob. | DTD Acc. | Rob. | EuroSAT Acc. | Rob. | UCF101 Acc. | Rob. | Avg. Acc. | Rob. |
|---|---|---|---|---|---|---|---|---|---|---|---|---|---|---|---|---|---|---|---|
| CW | CLIP | 85.9 | 22.7 | 83.5 | 7.7 | 55.7 | 6.6 | 61.7 | 5.8 | 15.7 | 8.8 | 40.4 | 16.3 | 23.7 | 15.4 | 58.9 | 11.3 | 53.2 | 11.8 |
| | Ensemble | 92.1 | 86.7 | 88.7 | 78.6 | 63.2 | 50.5 | 70.8 | 61.6 | 25.9 | 22.2 | 50.9 | 45.2 | 32.9 | 24.8 | 64.6 | 55.3 | 61.1 | 53.1 |
| | TPT | 94.1 | 77.2 | 87.4 | 66.9 | 66.5 | 43.2 | 66.1 | 49.9 | 23.4 | 16.1 | 45.9 | 30.7 | 42.6 | 13.5 | 67.9 | 44.1 | 61.7 | 42.7 |
| | C-TPT | 93.9 | 76.0 | 88.2 | 62.1 | 65.8 | 39.6 | 69.6 | 47.5 | 23.9 | 15.1 | 45.9 | 30.2 | 42.3 | 11.5 | 65.6 | 40.3 | 61.9 | 40.2 |
| | MTA | 94.3 | 79.2 | 88.0 | 67.8 | 67.7 | 43.4 | 65.0 | 48.4 | 24.0 | 16.1 | 46.5 | 30.6 | 42.5 | 19.3 | 67.5 | 44.6 | 61.9 | 43.6 |
| | R-TPT | 93.7 | 88.1 | 87.2 | 74.4 | 67.0 | 50.7 | 68.7 | 55.7 | 23.9 | 20.1 | 46.4 | 39.6 | 34.7 | 24.8 | 67.2 | 56.2 | 61.1 | 51.2 |
| | RITA | 93.8 | 87.8 | 89.8 | 78.7 | 64.2 | 52.6 | 71.6 | 62.0 | 26.2 | 22.9 | 51.5 | 46.2 | 33.4 | 25.2 | 65.5 | 56.8 | 62.0 | 54.0 |
| DI | CLIP | 85.9 | 23.0 | 83.5 | 4.4 | 55.7 | 0.6 | 61.7 | 1.9 | 15.7 | 0.0 | 40.4 | 6.1 | 23.7 | 0.0 | 58.9 | 3.2 | 53.2 | 4.9 |
| | Ensemble | 92.1 | 84.3 | 88.7 | 69.1 | 63.2 | 39.1 | 70.8 | 52.7 | 25.9 | 15.3 | 50.9 | 39.7 | 32.9 | 17.6 | 64.6 | 47.6 | 61.1 | 45.7 |
| | TPT | 94.1 | 80.7 | 87.4 | 65.2 | 66.5 | 38.3 | 66.1 | 49.7 | 23.4 | 13.5 | 45.9 | 30.3 | 42.6 | 7.4 | 67.9 | 40.5 | 61.7 | 40.7 |
| | C-TPT | 93.9 | 79.5 | 88.2 | 59.5 | 65.8 | 34.2 | 69.6 | 47.3 | 23.9 | 11.6 | 45.9 | 29.1 | 42.3 | 7.4 | 65.6 | 37.0 | 61.9 | 38.2 |
| | MTA | 94.3 | 82.6 | 88.0 | 65.6 | 67.7 | 39.5 | 65.0 | 48.0 | 24.0 | 13.5 | 46.5 | 30.9 | 42.5 | 14.7 | 67.5 | 41.9 | 61.9 | 42.1 |
| | R-TPT | 93.7 | 84.9 | 87.2 | 66.7 | 67.0 | 39.1 | 68.7 | 48.1 | 23.9 | 14.1 | 46.4 | 35.6 | 34.7 | 18.0 | 67.2 | 47.1 | 61.1 | 44.2 |
| | RITA | 93.8 | 84.7 | 89.8 | 69.4 | 64.2 | 39.9 | 71.6 | 53.4 | 26.2 | 15.9 | 51.5 | 39.8 | 33.4 | 18.4 | 65.5 | 47.7 | 62.0 | 46.2 |

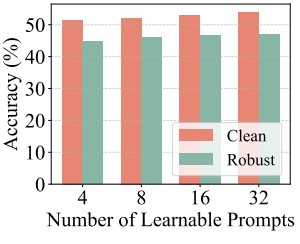
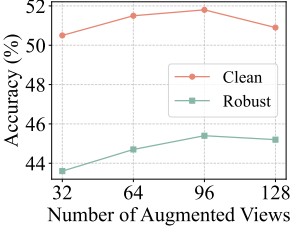

*(a)* Prompt Number  *(b)* Augmentation View

*Figure 3.* Ablation studies of key hyperparameters in DTD dataset using ViT-B/16 with $\epsilon = 1.0$. (a) Performance variation with respect to the number of learnable prompts. (b) Impact of the number of augmentation views during inference.

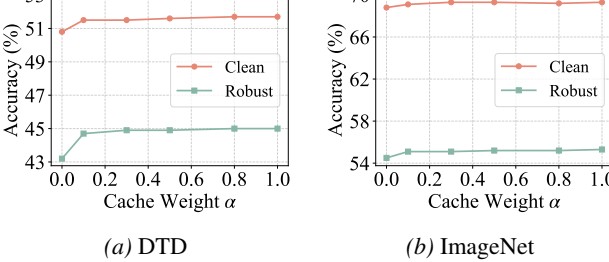

*(a)* DTD  *(b)* ImageNet

*Figure 4.* Sensitivity analysis of the cache integration coefficient $\alpha$. Classification accuracy (%) on (a) DTD and (b) ImageNet is evaluated across $\alpha$ values using ViT-B/16 with $\epsilon = 1.0$.

achieving a leading average accuracy of 54.0% under CW attacks. Under the more challenging DI attack, RITA maintains its advantage with 46.2% accuracy, surpassing R-TPT by 2.0%. These results suggest that RITA enhances the intrinsic robustness of VLMs against diverse adversarial threats via its effective test-time adaptation mechanism.

### 4.3. Ablation Study

**Number of learnable prompts and augmented views.** As illustrated in Figure 3, we conduct sensitivity analyses on the DTD dataset regarding the number of learnable prompts and augmentation views. Figure 3(a) shows that increasing the number of learnable prompts from 4 to 32 leads to steady performance gains across both settings. Figure 3(b) assesses the sensitivity of augmentation view scales during inference, where adversarial accuracy peaks at 96 views. Notably, scaling to 128 views results in a slight performance decline, a trend consistently observed in noise-free environments as well. Consequently, selecting a moderate number of augmentation views enables a superior trade-off between efficiency and accuracy.

**Contribution coefficient of dynamic cache.** We investigate the sensitivity of RITA to the integration coefficient $\alpha$ during inference on the DTD and ImageNet datasets, as shown in Figure 4. Experimental results confirm that the cache mechanism effectively boosts performance. On both datasets, increasing $\alpha$ from 0 to 0.1 yields a significant accuracy gain, followed by a slight upward trend that gradually stabilizes as $\alpha$ further scales. These results indicate that the visual priors stored in the cache not only play a crucial role in correcting predictions under adversarial settings, but also offer effective semantic complementarity during the inference stage. Ablation study of the dynamic cache is provided in Appendix D.1.

**Different perturbation budgets.** To assess the resilience of RITA under various attack intensities, we conduct ablation studies on all fine-grained datasets and ImageNet dataset, with the results illustrated in Figure 5. As the perturbation budget $\epsilon$ increases from 1 to 4, the accuracy declines as expected, indicating the generation of more potent perturbations. Simultaneously, we observe that increasing the number of TTA steps generally yields additional and consistent defensive gains across evaluations.

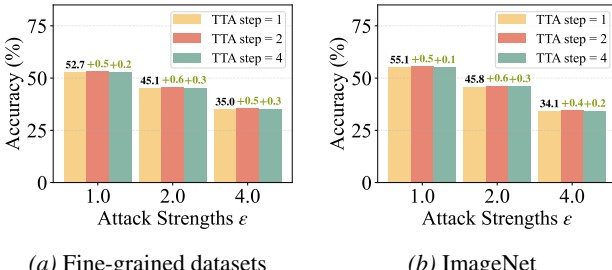

*(a)* Fine-grained datasets      *(b)* ImageNet

*Figure 5.* Adversarial robustness (%) under varying perturbation budgets and TTA steps on (a) fine-grained datasets and (b) ImageNet. Robust accuracy is evaluated using ViT-B/16 under $\epsilon \in \{1.0, 2.0, 4.0\}$ and TTA steps $\in \{1, 2, 4\}$. +x.x indicates the increment relative to the case where TTA step = 1.

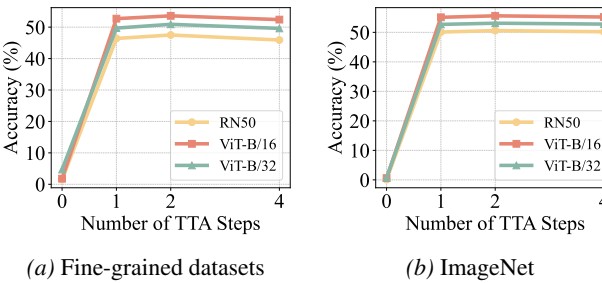

*(a)* Fine-grained datasets      *(b)* ImageNet

*Figure 6.* Evolution of adversarial robustness with respect to TTA steps across model architectures, where (a) presents the average robust accuracy over fine-grained datasets and (b) displays results on ImageNet. TTA step = 0 denotes the CLIP baseline.

**Number of TTA steps.** We report the average robust accuracy across three different backbones on all fine-grained datasets as shown in Figure 6(a) and the ImageNet dataset as illustrated in Figure 6(b). The experimental results indicate that all models reach their performance peak at step 2, while the accuracy slightly declines when the steps are increased to 4 across all evaluated datasets. This observation justifies our choice of a small number of iterations, *e.g.*, TTA step = 1, which simultaneously ensures superior robustness and high computational efficiency for real-time inference.

**Inference efficiency analysis.** Table 4 presents the average comparison of running time, clean accuracy, and robust accuracy across various methods on fine-grained datasets. The empirical results indicate that RITA achieves superior classification performance while maintaining highly competitive inference efficiency compared with existing adaptation methods. Compared to MTA, which exhibits the fastest inference speed, RITA maintains a substantial lead of 11.8% in robust accuracy. These observations demonstrate that RITA strikes an excellent balance between computational cost and model robustness, delivering more resilient and precise inference results with minimal additional time latency.

**Semantic alignment analysis.** As shown in Figure 7, we visualize semantic alignment quality via KDE curves of the

*Table 4.* Comparison of running time on fine-grained datasets using ViT-B/16 with $\epsilon = 1.0$.

| Model | Running Time | Accuracy | |
| --- | --- | --- | --- |
| | | Clean | Robust |
| TPT | 1.52s/image | 61.7 | 39.3 |
| C-TPT | 1.64s/image | 61.9 | 35.8 |
| MTA | 1.20s/image | 61.9 | 40.9 |
| R-TPT | 1.70s/image | 61.1 | 50.5 |
| RITA | 1.76s/image | **62.0** | **52.7** |

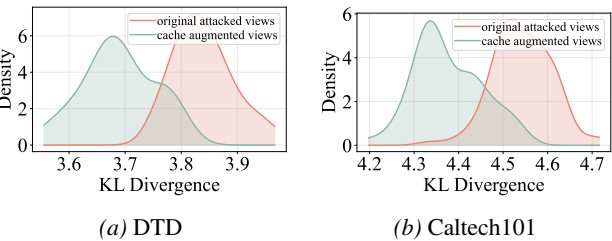

*(a)* DTD      *(b)* Caltech101

*Figure 7.* Comparison of KL divergence distributions for semantic alignment between original visual features and augmented features from the cache mechanism. Kernel Density Estimation (KDE) curves are presented for (a) DTD and (b) Caltech101 datasets. Lower KL values signify more deterministic vision-text alignment.

Kullback-Leibler (KL) divergence. Specifically, we measure the divergence between the class-conditional distributions over text prototypes and the ideal one-hot targets. On both DTD and Caltech101 datasets, the curves for augmented features exhibit a pronounced leftward shift and higher concentration in the low-value region compared to the original features. This demonstrates that our cache mechanism effectively rectifies adversarial semantic biases, establishing more deterministic vision-text associations. Extended analysis is provided in Appendix D.2.

## Conclusion & Limitations

In this work, we propose RITA, a robust test-time prompt adaptation framework that enhances the adversarial robustness of VLMs without requiring retraining or access to labeled data. By modeling augmented visual features and prompt-induced textual prototypes as distributions and aligning them via optimal transport, RITA corrects cross-modal semantic misalignment caused by adversarial perturbations. Furthermore, a dynamic cache mechanism progressively aggregates reliable semantic cues from the test stream to refine alignment online. Extensive experiments across diverse benchmarks, attack types, and model backbones demonstrate that RITA consistently improves adversarial robustness while preserving competitive performance on clean data. Our results suggest that distribution-level alignment is a principled and effective paradigm for robust inference in large pre-trained VLMs.

While RITA demonstrates strong image classification performance, its extension to generative tasks like image captioning remains for future exploration. We believe our distribution-level alignment provides a foundation for adaptation in these broader multimodal scenarios.

## Impact Statement

This work improves the reliability and adversarial robustness of pre-trained VLMs, which is important for their deployment in safety-sensitive applications. The proposed method operates entirely at test time, without modifying model parameters or requiring additional training data, making it lightweight and easy to integrate into existing systems. Nevertheless, it does not eliminate all security risks, and adversarial attacks may continue to evolve. Future work should further investigate robust inference-time defenses and evaluate potential failure modes and misuse risks.

## Acknowledgement

This research is supported by the National Natural Science Foundation of China (No. 62576330) and the National Natural Science Foundation of Anhui (No.2508085MF143).

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

# A. Proof of Theorem 1

**Theorem 3.1** (Decomposition of Alignment Objective). *The Optimal Transport objective ($\mathcal{L}_{\mathrm{OT}}$) imposes a stricter bound by decomposing into a centroid alignment term and a structural variance penalty:*

$$\underbrace{W_2^2(\mathbb{P}_t, \mathbb{Q}_k)}_{\mathcal{L}_{\mathrm{OT}} \text{ (RITA)}} \approx \underbrace{\|\boldsymbol{\mu}_x - \boldsymbol{\mu}_z\|^2}_{\mathcal{L}_{\mathrm{mean}} \text{ (TPT-equivalent)}} + \underbrace{\mathfrak{B}^2(\boldsymbol{\Sigma}_x, \boldsymbol{\Sigma}_z)}_{\mathcal{R}_{\mathrm{var}} \text{ (Structural Penalty)}}, \tag{13}$$

*where $\mathfrak{B}^2(\mathbf{A}, \mathbf{B}) = \mathrm{Tr}(\mathbf{A} + \mathbf{B} - 2(\mathbf{A}^{1/2}\mathbf{B}\mathbf{A}^{1/2})^{1/2})$ is the Bures metric, quantifying the geometric mismatch between covariances.*

*Proof.* Let $X \sim \mu$ and $Z \sim \nu$ be random vectors in $\mathbb{R}^d$ with mean vectors $\boldsymbol{\mu}_x, \boldsymbol{\mu}_z$ and covariance matrices $\boldsymbol{\Sigma}_x, \boldsymbol{\Sigma}_z$, respectively. The 2-Wasserstein distance is defined as the minimum expected transport cost over all valid joint couplings $\pi \in \Pi(\mu, \nu)$:

$$W_2^2(\mu, \nu) = \inf_{\pi \in \Pi(\mu,\nu)} \mathbb{E}_{(X,Z)\sim\pi}\left[\|X - Z\|^2\right]. \tag{14}$$

We first expand the squared Euclidean cost by centering the variables around their respective means. Let $\tilde{X} = X - \boldsymbol{\mu}_x$ and $\tilde{Z} = Z - \boldsymbol{\mu}_z$. The cost function can be rewritten as:

$$\begin{aligned} \|X - Z\|^2 &= \|(\tilde{X} - \tilde{Z}) + (\boldsymbol{\mu}_x - \boldsymbol{\mu}_z)\|^2 \\ &= \|\boldsymbol{\mu}_x - \boldsymbol{\mu}_z\|^2 + \|\tilde{X}\|^2 + \|\tilde{Z}\|^2 + 2(\boldsymbol{\mu}_x - \boldsymbol{\mu}_z)^\top(\tilde{X} - \tilde{Z}) - 2\tilde{X}^\top\tilde{Z}. \end{aligned} \tag{15}$$

Taking the expectation $\mathbb{E}_\pi[\cdot]$, the linear terms vanish because the variables are centered (i.e., $\mathbb{E}[\tilde{X}] = \mathbb{E}[\tilde{Z}] = 0$). Utilizing the identity $\mathbb{E}[\|\tilde{X}\|^2] = \mathrm{Tr}(\boldsymbol{\Sigma}_x)$, the expected cost simplifies to:

$$\mathbb{E}_\pi\left[\|X - Z\|^2\right] = \|\boldsymbol{\mu}_x - \boldsymbol{\mu}_z\|^2 + \mathrm{Tr}(\boldsymbol{\Sigma}_x) + \mathrm{Tr}(\boldsymbol{\Sigma}_z) - 2\mathbb{E}_\pi\left[\tilde{X}^\top\tilde{Z}\right]. \tag{16}$$

To find $W_2^2$, we minimize Eq. (16) over the coupling $\pi$. Since the mean difference and trace terms are constants independent of $\pi$, the problem reduces to maximizing the correlation term $\mathbb{E}_\pi[\tilde{X}^\top\tilde{Z}]$. For the family of elliptical distributions (e.g., Gaussians), it is a known result (Gelbrich, 1990) that the optimal coupling yields:

$$\sup_{\pi \in \Pi} \mathbb{E}_\pi\left[\tilde{X}^\top\tilde{Z}\right] = \mathrm{Tr}\left((\boldsymbol{\Sigma}_x^{1/2}\boldsymbol{\Sigma}_z\boldsymbol{\Sigma}_x^{1/2})^{1/2}\right). \tag{17}$$

Substituting this optimal correlation back into Eq. (16), we obtain:

$$\begin{aligned} W_2^2(\mu, \nu) &= \|\boldsymbol{\mu}_x - \boldsymbol{\mu}_z\|^2 + \mathrm{Tr}(\boldsymbol{\Sigma}_x) + \mathrm{Tr}(\boldsymbol{\Sigma}_z) - 2\,\mathrm{Tr}\left((\boldsymbol{\Sigma}_x^{1/2}\boldsymbol{\Sigma}_z\boldsymbol{\Sigma}_x^{1/2})^{1/2}\right) \\ &= \underbrace{\|\boldsymbol{\mu}_x - \boldsymbol{\mu}_z\|^2}_{\mathcal{L}_{\mathrm{mean}}} + \underbrace{\mathrm{Tr}\left(\boldsymbol{\Sigma}_x + \boldsymbol{\Sigma}_z - 2(\boldsymbol{\Sigma}_x^{1/2}\boldsymbol{\Sigma}_z\boldsymbol{\Sigma}_x^{1/2})^{1/2}\right)}_{\mathfrak{B}^2(\boldsymbol{\Sigma}_x, \boldsymbol{\Sigma}_z)}. \end{aligned} \tag{18}$$

The first term corresponds to the centroid distance ($\mathcal{L}_{\mathrm{mean}}$), and the second term is the squared Bures metric ($\mathfrak{B}^2$), representing the structural variance cost. For general distributions, this expression serves as a tight lower bound, confirming that minimizing $W_2^2$ inherently constrains both the first-order (mean) and second-order (variance) geometric moments. $\square$ $\square$

*Table 5.* Results (%) of various adaptation methods on ImageNet and ImageNet-OOD datasets with $\epsilon = 1.0$. OOD Avg. refers to the average results among four ImageNet-OOD datasets.

| | Method | ImageNet | | ImageNet-A | | ImageNet-V2 | | ImageNet-R | | ImageNet-S | | OOD Avg. | |
|---|---|---|---|---|---|---|---|---|---|---|---|---|---|
| | | Acc. | Rob. | Acc. | Rob. | Acc. | Rob. | Acc. | Rob. | Acc. | Rob. | Acc. | Rob. |
| ViT-B/32 | CLIP | 62.0 | 0.7 | 29.5 | 0.1 | 54.7 | 1.5 | 66.2 | 6.9 | 40.8 | 4.5 | 47.8 | 3.2 |
| | Ensemble | 64.4 | 52.3 | 34.1 | 21.4 | 58.1 | 45.6 | 64.5 | 55.2 | 39.2 | 31.2 | 49.0 | 38.3 |
| | TPT | 63.6 | 36.6 | 34.5 | 9.3 | 56.9 | 30.4 | 69.1 | 49.4 | 41.6 | 30.4 | 50.5 | 31.2 |
| | C-TPT | 63.5 | 33.4 | 30.5 | 7.6 | 55.9 | 27.4 | 67.0 | 45.0 | 41.8 | 30.1 | 48.8 | 28.7 |
| | MTA | **64.9** | 40.1 | **37.7** | 11.1 | 58.3 | 33.2 | **70.3** | 52.3 | **43.4** | 31.5 | **52.4** | 32.0 |
| | R-TPT | 64.4 | 52.1 | 36.9 | 21.9 | 58.0 | 45.5 | 67.5 | 55.8 | 41.7 | 31.2 | 51.0 | 38.6 |
| | RITA | 64.8 | **52.7** | 35.4 | **22.5** | 58.5 | 45.9 | 65.7 | **56.3** | 40.8 | **32.3** | 50.1 | **39.3** |
| ViT-B/16 | CLIP | 66.7 | 0.6 | 47.7 | 0.1 | 60.8 | 0.2 | 73.9 | 3.5 | 46.1 | 2.2 | 57.1 | 1.5 |
| | Ensemble | 68.8 | 54.4 | 55.8 | 33.6 | 62.8 | 47.4 | 72.9 | 62.7 | 46.5 | 35.1 | 59.5 | 44.7 |
| | TPT | 68.9 | 42.4 | 54.7 | 14.9 | **63.6** | 35.7 | **77.1** | 57.3 | 47.9 | 35.6 | 60.8 | 37.1 |
| | C-TPT | 68.1 | 38.0 | 49.7 | 11.3 | 61.9 | 31.4 | 74.8 | 51.9 | 47.2 | 34.4 | 58.4 | 3.4 |
| | MTA | 69.0 | 44.4 | **57.3** | 17.5 | 63.4 | 37.2 | 76.9 | 58.9 | **48.4** | 35.8 | **61.5** | 38.7 |
| | R-TPT | **69.1** | 54.4 | 57.2 | 34.7 | 63.5 | 48.0 | 75.5 | 63.7 | 47.7 | 36.5 | 60.9 | 45.7 |
| | RITA | **69.1** | **55.1** | 55.8 | **35.0** | 63.2 | **48.4** | 74.0 | **64.1** | 47.1 | **37.2** | 60.0 | **46.2** |

## B. Implementation Details

For fair comparison, all approaches use the same pre-trained CLIP backbone and identical AugMix-based test-time augmentations, without external data, foundation models, or additional knowledge. We report average accuracy on clean samples and adversarial accuracy under PGD attacks with different perturbation budgets for default. Adversarial examples are generated on the original CLIP model, reflecting a realistic threat model.

For most experiments, we adopt a default setting that utilizes class descriptors and sets the subset size $|\mathcal{S}| = 64$. However, to accommodate the distinct characteristics of specific benchmarks, we adjust these parameters for several datasets as follows: 1) For EuroSAT, we disable descriptors for the ViT-B/16 backbone and set $|\mathcal{S}| = 32$ for both ViT-B/16 and ViT-B/32; 2) For ImageNet-A and ImageNet-R, the size of subset $|\mathcal{S}|$ is 32; 3) For ImageNet-Sketch, descriptors are disabled, and the subset size $|\mathcal{S}|$ is 32. For all other datasets not explicitly mentioned, the default configuration is maintained. These refinements are designed to better align domain-specific semantic features with the test-time adaptation process.

## C. Additional Results

### C.1. Results on ImageNet and ImageNet-OOD datasets.

Table 5 presents the performance comparison across ImageNet and its four Out-of-Distribution (OOD) variants. The results demonstrate that RITA maintains a significant robustness advantage even when addressing large-scale general visual tasks. On the standard ImageNet, RITA achieves state-of-the-art or competitive accuracies across both architectures under both settings; for instance, it reaches 55.1% robustness on ViT-B/16, a substantial leap from the vanilla CLIP's 0.6%. RITA's performance is equally compelling on the more challenging OOD variants, where its average OOD robust accuracy reaches 39.3% and 46.2% in two backbones, outperforming other methods. These findings validate that RITA not only excels in fine-grained tasks but also provides robust protection against diverse adversarial threats in large-scale general scenarios and under various distribution shifts.

### C.2. Evaluation under enhanced adversarial attacks.

Table 6 presents the performance of various test-time adaptation methods across eight fine-grained datasets under a more stringent adversarial constraint, where $\epsilon$ is set to 4.0. The experimental results indicate that as the attack intensity increases, the accuracy of the vanilla CLIP drops nearly to zero, whereas RITA demonstrates exceptional interference resistance. Specifically, our method achieves state-of-the-art results across both backbones, reaching 32.9% and 35.0% in average robust accuracy, respectively. These findings further confirm that RITA maintains high classification precision while exhibiting remarkable predictive stability, thereby validating its substantial practical value in mitigating complex and intense adversarial risks in real-world scenarios.

*Table 6.* Results (%) of adaptation methods on fine-grained classification datasets with $\epsilon = 4.0$.

| | Method | Caltech101 Acc. | Rob. | Pets Acc. | Rob. | Cars Acc. | Rob. | Flower102 Acc. | Rob. | Aircraft Acc. | Rob. | DTD Acc. | Rob. | EuroSAT Acc. | Rob. | UCF101 Acc. | Rob. | Avg. Acc. | Rob. |
|---|---|---|---|---|---|---|---|---|---|---|---|---|---|---|---|---|---|---|---|
| ViT-B/32 | CLIP | 90.9 | 2.5 | 83.0 | 0.0 | 49.7 | 0.0 | 65.8 | 0.0 | 18.3 | 0.0 | 40.8 | 0.2 | 18.6 | 0.0 | 62.1 | 0.0 | 53.6 | 0.3 |
| | Ensemble | 91.6 | 73.7 | 85.0 | 47.0 | 57.8 | 18.5 | 67.4 | 34.8 | 20.1 | 7.2 | 46.1 | 29.2 | 32.5 | 7.4 | 61.6 | 37.8 | 57.8 | 32.0 |
| | TPT | 91.4 | 60.0 | 84.1 | 30.3 | 62.9 | 16.4 | 63.8 | 28.5 | 19.0 | 3.9 | 42.2 | 19.9 | 35.1 | 6.5 | 62.3 | 21.8 | 57.6 | 23.4 |
| | C-TPT | 91.8 | 53.7 | 84.9 | 21.5 | 60.8 | 9.2 | 65.9 | 22.1 | 17.7 | 2.4 | 44.3 | 16.4 | 34.7 | 5.7 | 62.6 | 17.8 | 57.8 | 18.6 |
| | MTA | 91.8 | 73.9 | 85.8 | 45.8 | 64.1 | 19.2 | 64.8 | 34.2 | 20.4 | 6.1 | 44.0 | 22.0 | 34.5 | 5.3 | 63.6 | 33.1 | 58.6 | 29.9 |
| | R-TPT | 90.6 | 74.6 | 84.5 | 44.5 | 63.1 | 20.5 | 62.6 | 34.1 | 19.1 | 6.6 | 42.1 | 27.3 | 32.0 | 7.1 | 62.8 | 37.1 | 57.1 | 31.4 |
| | RITA | 92.3 | 74.8 | 85.9 | 47.4 | 59.6 | 20.1 | 68.7 | 35.6 | 20.2 | 7.7 | 46.2 | 30.2 | 33.4 | 8.5 | 62.8 | 39.1 | 58.6 | 32.9 |
| ViT-B/16 | CLIP | 85.9 | 0.7 | 83.5 | 0.0 | 55.7 | 0.0 | 61.7 | 0.0 | 15.7 | 0.0 | 40.4 | 0.0 | 23.7 | 0.0 | 58.9 | 0.0 | 53.2 | 0.1 |
| | Ensemble | 92.1 | 76.8 | 88.7 | 47.9 | 63.2 | 22.4 | 70.8 | 37.6 | 25.9 | 10.2 | 50.9 | 33.2 | 32.9 | 6.6 | 64.6 | 35.6 | 61.1 | 33.8 |
| | TPT | 94.1 | 60.6 | 87.4 | 31.0 | 66.5 | 13.8 | 66.1 | 23.7 | 23.4 | 4.4 | 45.9 | 17.4 | 42.6 | 4.6 | 67.9 | 20.3 | 61.7 | 21.9 |
| | C-TPT | 93.9 | 49.6 | 88.2 | 21.1 | 65.8 | 9.2 | 69.6 | 17.2 | 23.9 | 2.0 | 45.9 | 12.7 | 42.3 | 5.2 | 65.6 | 14.2 | 61.9 | 16.4 |
| | MTA | 94.3 | 73.6 | 88.0 | 51.2 | 67.7 | 25.7 | 65.0 | 31.7 | 24.0 | 7.4 | 46.5 | 21.5 | 42.5 | 6.5 | 67.5 | 30.9 | 61.9 | 31.0 |
| | R-TPT | 93.7 | 78.3 | 87.2 | 45.6 | 67.0 | 23.9 | 68.7 | 34.8 | 23.9 | 10.5 | 46.4 | 30.4 | 34.7 | 6.3 | 67.2 | 35.2 | 61.1 | 33.1 |
| | RITA | 93.8 | 78.5 | 89.8 | 48.1 | 64.2 | 24.2 | 71.6 | 38.4 | 26.2 | 11.3 | 51.5 | 34.6 | 33.4 | 7.9 | 65.5 | 37.2 | 62.0 | 35.0 |

*Table 7.* Results (%) of adaptation methods on fine-grained classification datasets using RN50 with $\epsilon$ set to 1.0.

| Method | Caltech101 Acc. | Rob. | Pets Acc. | Rob. | Cars Acc. | Rob. | Flower102 Acc. | Rob. | Aircraft Acc. | Rob. | DTD Acc. | Rob. | EuroSAT Acc. | Rob. | UCF101 Acc. | Rob. | Avg. Acc. | Rob. |
|---|---|---|---|---|---|---|---|---|---|---|---|---|---|---|---|---|---|---|
| CLIP | 84.9 | 2.6 | 83.5 | 0.0 | 53.7 | 0.0 | 61.7 | 0.0 | 14.7 | 0.0 | 40.4 | 0.8 | 18.7 | 0.0 | 57.5 | 0.0 | 51.8 | 0.4 |
| Ensemble | 84.6 | 78.2 | 85.1 | 75.4 | 52.6 | 38.4 | 65.4 | 56.2 | 15.7 | 11.8 | 43.1 | 38.2 | 22.0 | 14.6 | 56.3 | 49.5 | 53.1 | 45.2 |
| TPT | 85.1 | 7.0 | 84.7 | 0.1 | 54.4 | 0.0 | 62.1 | 0.0 | 15.3 | 5.2 | 42.4 | 4.3 | 22.4 | 0.0 | 60.2 | 0.3 | 53.3 | 2.1 |
| C-TPT | 84.8 | 3.7 | 83.6 | 0.0 | 55.6 | 0.0 | 64.8 | 0.0 | 16.7 | 7.2 | 41.5 | 1.3 | 22.2 | 0.0 | 60.1 | 0.1 | 53.6 | 1.5 |
| MTA | 85.3 | 65.9 | 84.8 | 59.8 | 55.7 | 17.8 | 61.0 | 31.5 | 15.9 | 10.3 | 40.3 | 18.8 | 22.5 | 1.6 | 60.6 | 31.3 | 53.2 | 29.6 |
| R-TPT | 86.7 | 78.2 | 84.6 | 74.2 | 56.1 | 38.6 | 60.6 | 51.9 | 16.4 | 11.8 | 41.3 | 33.5 | 21.2 | 15.1 | 59.5 | 49.2 | 53.3 | 43.0 |
| RITA | 85.7 | 79.3 | 86.3 | 77.4 | 54.2 | 39.9 | 66.2 | 56.9 | 16.9 | 12.4 | 43.8 | 39.5 | 20.2 | 15.9 | 58.8 | 50.0 | 54.0 | 46.4 |

*Table 8.* Robust accuracy (%) on fine-grained classification datasets under different attacks using ViT-B/16 with $\epsilon = 1.0$.

| | Method | Caltech101 | Pets | Cars | Flower102 | Aircraft | DTD | EuroSAT | UCF101 | Avg. |
|---|---|---|---|---|---|---|---|---|---|---|
| adaptive | CLIP | 30.4 | 4.9 | 7.2 | 3.2 | 0.2 | 2.4 | 0.0 | 0.8 | 6.1 |
| | R-TPT | 87.3 | 76.3 | 55.3 | 60.8 | 17.4 | 35.3 | 20.5 | 52.2 | 50.6 |
| | RITA | 88.2 | 79.5 | 57.8 | 66.2 | 19.1 | 37.6 | 23.8 | 54.8 | 53.4 |
| AA | CLIP | 13.2 | 4.9 | 0.3 | 2.6 | 0.0 | 0.0 | 0.0 | 4.6 | 3.2 |
| | R-TPT | 87.9 | 78.0 | 51.4 | 59.4 | 20.3 | 41.5 | 24.2 | 58.2 | 52.6 |
| | RITA | 89.8 | 82.0 | 53.7 | 61.8 | 22.6 | 47.7 | 26.0 | 59.9 | 55.4 |
| FGSM | CLIP | 6.2 | 2.4 | 0.5 | 0.0 | 0.0 | 0.4 | 0.0 | 1.8 | 1.4 |
| | R-TPT | 84.8 | 73.6 | 43.6 | 54.3 | 19.9 | 36.2 | 23.1 | 50.3 | 48.2 |
| | RITA | 85.9 | 74.5 | 44.2 | 59.4 | 21.7 | 42.1 | 24.4 | 51.8 | 50.5 |

## C.3. Analysis on an alternative CLIP backbone.

In Table 7, we further evaluate the performance of RITA using RN50 as the vision backbone to verify its generalizability across different architectures. The experimental results demonstrate that RITA exhibits superior robustness across all eight fine-grained classification datasets. Notably, despite RN50 having a relatively weaker baseline representation capability compared to the ViT series, RITA consistently outperforms other adaptation methods, while maintaining high clean accuracy. These findings provide strong evidence that our method delivers cross-architecture robustness gains and effectively mitigates the vulnerability of diverse vision encoders to adversarial attacks.

## C.4. Robustness evaluation under other attacks.

We conduct experiments under various adversarial attack protocols, including adaptive attack that is aware of augmentation strategies, AutoAttack (AA), and FGSM. As reported in Table 8, RITA consistently achieves the highest robust accuracy

*Table 9.* Classification accuracy (%) on 10 datasets using EVA-CLIP and OpenCLIP backbones.

| Method | Caltech101 Acc. | Caltech101 Rob. | Pets Acc. | Pets Rob. | Cars Acc. | Cars Rob. | Flower102 Acc. | Flower102 Rob. | Aircraft Acc. | Aircraft Rob. | DTD Acc. | DTD Rob. | EuroSAT Acc. | EuroSAT Rob. | UCF101 Acc. | UCF101 Rob. | SUN397 Acc. | SUN397 Rob. | Food101 Acc. | Food101 Rob. | Avg. Acc. | Avg. Rob. |
|---|---|---|---|---|---|---|---|---|---|---|---|---|---|---|---|---|---|---|---|---|---|---|
| OpenCLIP | 91.3 | 12.3 | 89.2 | 1.2 | 75.7 | 2.9 | 66.9 | 0.2 | 17.7 | 0.0 | 51.3 | 3.1 | 50.1 | 0.4 | 67.3 | 0.1 | 69.6 | 1.9 | 85.9 | 1.4 | 66.5 | 2.4 |
| + RITA | 92.4 | 89.9 | 91.7 | 78.2 | 78.4 | 50.2 | 73.2 | 62.4 | 25.1 | 19.3 | 55.9 | 46.3 | 51.5 | 29.9 | 69.6 | 53.5 | 75.2 | 54.0 | 88.3 | 60.7 | 70.1 | 54.4 |
| EVA-CLIP | 86.3 | 5.2 | 92.2 | 0.3 | 78.6 | 4.5 | 75.9 | 1.2 | 24.8 | 0.0 | 53.1 | 1.7 | 67.0 | 0.5 | 63.2 | 0.0 | 79.7 | 4.2 | 89.4 | 0.9 | 71.0 | 1.6 |
| + RITA | 87.1 | 84.6 | 93.1 | 80.4 | 79.5 | 48.2 | 78.6 | 64.8 | 27.5 | 17.5 | 56.2 | 41.4 | 69.8 | 38.2 | 64.9 | 50.8 | 82.3 | 59.2 | 91.2 | 62.4 | 73.0 | 54.8 |

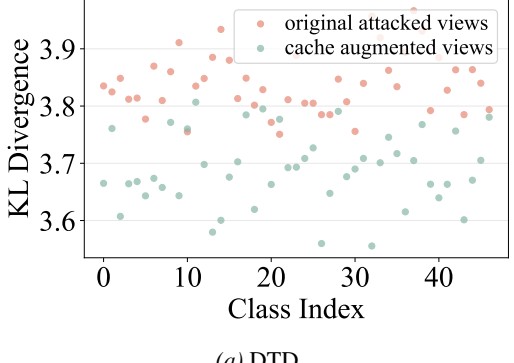

*(a)* DTD

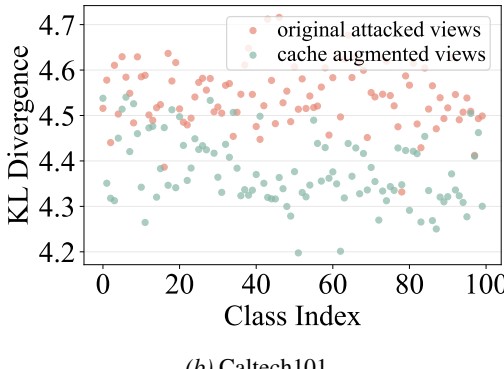

*(b)* Caltech101

*Figure 8.* KL divergence per class on (a) DTD and (b) Caltech101. Lower KL values signify superior vision-text alignment.

across all eight datasets and all attack types compared to the CLIP and R-TPT baselines. Specifically, under the more rigorous AutoAttack, RITA maintains an average robust accuracy of 55.4%, outperforming R-TPT by 2.8%. Notably, RITA shows significant gains on challenging datasets like DTD and EuroSAT across all attack settings. These results demonstrate that the defensive capability of RITA is not tailored to a specific attack but generalizes well to diverse adversarial threats, confirming its potential for securing VLMs in various hostile environments.

### C.5. Generalization to Other VLM Backbones.

To further validate the architectural agnosticity of RITA, we extend our evaluation to other VLMs, including OpenCLIP and EVA-CLIP. As shown in Table 9, we expanded RITA to 10 datasets by incorporating SUN397 and Food101. The results consistently demonstrate that RITA significantly boosts adversarial robustness across all architectures. For instance, when applied to OpenCLIP, RITA improves the average robust accuracy from 2.4% to 54.4%. On the newly added SUN397 and Food101 datasets, RITA achieves substantial gains, reaching robust accuracies of 54.0% and 60.7% for OpenCLIP, and 59.2% and 62.4% for EVA-CLIP, respectively. These findings underscore that the robustness gains of RITA are consistent across diverse VLM backbones and broader dataset distributions, reinforcing its effectiveness as a general test-time adaptation framework.

## D. Verification of the importance of the cache mechanism

### D.1. Ablation study of the dynamic cache.

In Table 10, we conduct an ablation study to specifically validate the significance of the cache mechanism $d_{cOT}$ within the RITA framework. The results indicate that while utilizing the cache mechanism in isolation yields lower standard accuracy (39.7% on fine-grained and 20.2% on ImageNet) due to the absence of real-time alignment, it consistently outperforms the vanilla CLIP baseline in robustness metrics, achieving 27.1% and 29.6% respectively, compared to CLIP's 4.7% and 3.2%. This evidence underscores that the historical priors preserved in the cache serve as an essential reference for stabilizing predictions under adversarial perturbations. Most importantly, the synergy between the cache mechanism and the optimal transport module $d_{OT}$ leads to peak performance across all metrics, notably boosting fine-grained robustness from 49.4% to 55.0%. This further demonstrates that the cache provides critical semantic supplementation to the multimodal alignment, establishing a more robust and historically-aware inference framework.

*Table 10.* Main component analysis (%) on fine-grained datasets and ImageNet dataset using ViT-B/32 with $\epsilon = 1.0$.

| $d_{\mathrm{OT}}$ | $d_{\mathrm{cOT}}$ | Fine-grained | | ImageNet | |
|---|---|---|---|---|---|
| | | Acc. | Rob. | Acc. | Rob. |
| ✗ | ✗ | 53.6 | 4.7 | 47.8 | 3.2 |
| ✔ | ✗ | 57.0 | 49.4 | 63.9 | 52.3 |
| ✗ | ✔ | 39.7 | 27.1 | 20.2 | 29.6 |
| ✔ | ✔ | **57.9** | **55.0** | **64.8** | **52.7** |

---

**Algorithm 1** RITA: Robust test-tIme prompT Adaptation

---

1: **Input:** Test stream $\{\hat{x}_t\}$, encoders $\Phi_{\mathrm{img}}, \Phi_{\mathrm{text}}$, text prompts $\{z_k^{(m)}\}_{m=1}^M$, entropy threshold $\gamma$, max cache size per class $N_k$, cache weight $\alpha$.
2: **Initialize:** Empty cache $\{\hat{X}_k\}_{k=1}^K \leftarrow \emptyset$; extract text features $\mathbf{z}_k^m = \Phi_{\mathrm{text}}(z_k^{(m)})$ and construct text distributions $\mathbb{Q}_k = \frac{1}{M}\sum_{m=1}^M \delta_{\mathbf{z}_k^m}$.
3: **for** each test image $\hat{x}_t$ **do**
4:     Generate $N$ augmented views $\{\hat{x}_t^n\}_{n=1}^N$, extract visual features $\mathbf{x}_t^n = \Phi_{\mathrm{img}}(\hat{x}_t^n)$, and construct visual distribution $\mathbb{P}_t = \frac{1}{N}\sum_{n=1}^N \delta_{\mathbf{x}_t^n}$.
5:     **Cache Update:** For each view $\hat{x}_t^n$ with entropy $H(p_t^n) < \gamma$, get pseudo-label $\hat{k} = \operatorname*{argmax}_k p_t^n(k)$. Add $\mathbf{x}_t^n$ to $\hat{X}_{\hat{k}}$ (if $|\hat{X}_{\hat{k}}| \geq N_k$ and $H(p_t^n)$ is lower, replace the max-entropy sample).
6:     **for** class $k = 1$ **to** $K$ **do**
7:         Compute global OT distance $d_{OT}(\mathbb{P}_t, \mathbb{Q}_k; C_{t,k})$ where $C_{t,k}(n, m) = 1 - \cos(\mathbf{x}_t^n, \mathbf{z}_k^m)$.
8:         **If** $\hat{X}_k \neq \emptyset$, align features $\tilde{X}_k = \hat{X}_k W_k^*$ to build cache dist $\tilde{\mathbb{Q}}_k$, and calculate cache OT distance $d_{cOT}(\mathbb{P}_t, \tilde{\mathbb{Q}}_k; \tilde{C}_{t,k})$
9:         **Else** $d_{cOT} = 0$.
10:    **end for**
11:    **Output:** Predicted label $\hat{y} = \operatorname*{argmin}_{k \in [K]} \left( d_{OT}(\mathbb{P}_t, \mathbb{Q}_k; C_{t,k}) + \alpha d_{cOT}(\mathbb{P}_t, \tilde{\mathbb{Q}}_k; \tilde{C}_{t,k}) \right)$.
12: **end for**

---

### D.2. Semantic alignment analysis of the cache mechanism.

To demonstrate the effectiveness of the cache mechanism, we extract the augmented view features from the DTD and Caltech101 datasets and compare them with the original unaugmented visual features from a "vision-text" alignment perspective. Specifically, for each class, we first compute the class-conditional assignment distribution over all text prototypes, which reflects how the visual features of each class are semantically aligned with the textual features. Subsequently, we measure the KL divergence between this empirical distribution and an ideal one-hot target distribution that assigns all probability mass to the ground-truth class. A lower KL value indicates stronger vision-text alignment.

As illustrated in Figure 8, we present the scatter plots of KL divergence for all classes in DTD and Caltech101. Each point represents the vision-text alignment quality of a specific category, where red dots denote original adversarial views and green dots represent selected augmented views utilized by our cache mechanism. The KL divergence of original views exhibits high variance and remains at an elevated level, reflecting severe adversarial bias. Upon introducing the cache mechanism, the green dots show both a downward shift and reduced dispersion. This indicates that the augmented views effectively calibrate the corrupted feature representations by aggregating historical priors, pulling the model closer to the ideal one-hot distribution. Even for categories where the original KL divergence is particularly high, the augmented features still achieve significant alignment gains. This cross-category consistency provides a granular foundation for the superior robustness of the RITA framework when encountering diverse adversarial attacks.

## E. Algorithm for RITA

Algorithm 1 summarizes the RITA framework. RITA extracts features from augmented views, aligns visual-textual distributions via optimal transport, and maintains a dynamic cache for progressive refinement.

