# OpenReview forum: "Robustifying Vision-Language Models via Test-Time Prompt Adaptation"
_ICML.cc/2026/Conference — ICML 2026 regular_

### Official Review · Reviewer_9JQd · 2026-02-27

**Soundness:** 3
**Presentation:** 3
**Significance:** 3
**Originality:** 3
**Overall Recommendation:** 4
**Confidence:** 4

**Summary:**

This paper proposes RITA, a test-time prompt adaptation framework to improve the adversarial robustness of CLIP. Unlike prior methods that rely on sample-level confidence heuristics, RITA performs distribution-level alignment using optimal transport to match augmented visual features with textual prototypes, combined with a dynamic cache for online refinement. The authors claim significant robustness gains across multiple VLM backbones without sacrificing clean accuracy.

**Compliance With Llm Reviewing Policy:**

Affirmed.

**Final Justification:**

I would like to thank the authors for their thorough response. In the rebuttal, the authors directly addressed and effectively resolved all of my core concerns.

**Key Questions For Authors:**

1. The paper focuses on improving the robustness of CLIP. While this is a significant contribution, the experiments are all conducted on CLIP model only, without results on other CLIP-like models, e.g., OpenCLIP, EVA-CLIP, which may limit the contribution of proposed methods.

2. Incomplete experimental comparison. The paper omits closely related baselines such as TAPT and also lacks evaluation on standard benchmarks like Food101 and SUN397 that are widely used in prior work.

3. The PGD reference in the Implementation Details section is incorrect (it references CW). Moreover, the evaluated attack methods are too few. Including diverse attacks (e.g., AutoAttack, CW, FGSM) would better validate robustness claims.

**Limitations:**

Reference Weaknesses  and Questions

**Strengths And Weaknesses:**

Strengths

1. The manuscript is well-organized and readable.

2. The experimental setup is described in a detailed and thorough manner.

Weaknesses

1. The RITA computational cost may be very high.

---

> ### Author Rebuttal · Authors · 2026-03-31
>
> Dear reviewer 9JQd,
>
> Thank you for the insightful feedback! We have addressed your concerns below and incorporated all additions into the revision.
>
> ---
>
> **W1: Computational Cost**
>
> **A:** We break down the inference time for CLIP, TPT, R-TPT, and RITA, as shown in the table below. Compared to TTA methods like TPT and R-TPT, our unique OT computation and dynamic cache updating introduce minimal overhead.
>
> |Component|CLIP|TPT|R-TPT|RITA|
> |:---|:---:|:---:|:---:|:---:|
> |Data Augmentation|N/A|0.7932s|0.8433s|0.8411s|
> |Forward Pass|0.1043s|0.6882s|0.8210s|0.8324s|
> |OT Computation|N/A|N/A|N/A|0.0418s (2.37%)|
> |Cache Updating|N/A|N/A|N/A|0.0026s (0.15%)|
> |Loss|N/A|0.0389s|0.0406s|0.0453s|
> |**Total Time/Image**|**0.1043s**|**1.5203s**|**1.7049s**|**1.7632s**|
>
> ---
>
> **Q1: Generalization to Other VLM Backbones**
>
> **A:** We have conducted experiments with OpenCLIP and EVA-CLIP backbones on fine-grained datasets. As shown in the table below, RITA consistently improves robust accuracy across these different models, confirming that our approach generalizes well beyond CLIP.
>
> |Backbone|Method|Caltech101||Pets||Cars||Flower102||Aircraft||DTD||EuroSAT||UCF101||SUN397||Food101||Avg.||
> |:---|:---|:---:|:---:|:---:|:---:|:---:|:---:|:---:|:---:|:---:|:---:|:---:|:---:|:---:|:---:|:---:|:---:|:---:|:---:|:---:|:---:|:---:|:---:|
> |||**Acc.**|**Rob.**|**Acc.**|**Rob.**|**Acc.**|**Rob.**|**Acc.**|**Rob.**|**Acc.**|**Rob.**|**Acc.**|**Rob.**|**Acc.**|**Rob.**|**Acc.**|**Rob.**|**Acc.**|**Rob.**|**Acc.**|**Rob.**|**Acc.**|**Rob.**|
> |**OpenCLIP**|Zero-shot|91.3|12.3|89.2|1.2|75.7|2.9|66.9|0.2|17.7|0.0|51.3|3.1|50.1|0.4|67.3|0.1|69.6|1.9|85.9|1.4|66.5|2.4|
> ||**RITA**|**92.4**|**89.9**|**91.7**|**78.2**|**78.4**|**50.2**|**73.2**|**62.4**|**25.1**|**19.3**|**55.9**|**46.3**|**51.5**|**29.9**|**69.6**|**53.5**|**75.2**|**54.0**|**88.3**|**60.7**|**70.13**|**54.4**|
> |**EVA-CLIP**|Zero-shot|86.3|5.2|92.2|0.3|78.6|4.5|75.9|1.2|24.8|0.0|53.1|1.7|67.0|0.5|63.2|0.0|79.7|4.2|89.4|0.9|71.02|1.6|
> ||**RITA**|**87.1**|**84.6**|**93.1**|**80.4**|**79.5**|**48.2**|**78.6**|**64.8**|**27.5**|**17.5**|**56.2**|**41.4**|**69.8**|**38.2**|**64.9**|**50.8**|**82.3**|**59.2**|**91.2**|**62.4**|**73.02**|**54.8**|
>
> ---
>
> **Q2: Additional Baselines and Datasets**
>
> **A:** We have added Food101 and SUN397 to our evaluation and implemented TAPT under identical settings. The results below show that RITA outperforms TAPT in both clean accuracy and adversarial robustness.
>
> |Method|Caltech101||Pets||Cars||Flower102||Aircraft||DTD||EuroSAT||UCF101||SUN397||Food101||Avg.||
> |:---|:---:|:---:|:---:|:---:|:---:|:---:|:---:|:---:|:---:|:---:|:---:|:---:|:---:|:---:|:---:|:---:|:---:|:---:|:---:|:---:|:---:|:---:|
> ||**Acc.**|**Rob.**|**Acc.**|**Rob.**|**Acc.**|**Rob.**|**Acc.**|**Rob.**|**Acc.**|**Rob.**|**Acc.**|**Rob.**|**Acc.**|**Rob.**|**Acc.**|**Rob.**|**Acc.**|**Rob.**|**Acc.**|**Rob.**|**Acc.**|**Rob.**|
> |TAPT|89.9|84.1|85.5|76.9|61.7|46.8|65.0|58.8|24.7|16.4|42.6|41.2|29.5|19.3|64.4|53.6|65.3|54.7|84.9|57.1|61.4|50.9|
> |**RITA**|**93.8**|**88.5**|**89.8**|**77.3**|**64.2**|**47.1**|**71.6**|**61.3**|**26.2**|**19.2**|**51.5**|**44.7**|**33.4**|**27.6**|**65.5**|**55.8**|**70.8**|**56.6**|**86.7**|**58.3**|**65.4**|**53.6**|
>
> ---
>
> **Q3: Attack Diversity and Citation Mismatch**
>
> **A:** We have already included the results against the CW attack in **Table 3**. Following your suggestion, we have added evaluations under FGSM and AutoAttack. RITA consistently maintains robust defensive performance across diverse attack strategies. We have also updated the PGD citation in the revision.
>
> |Attack|Method|Caltech101 Rob.|Pets Rob.|Cars Rob.|Flower102 Rob.|Aircraft Rob.|DTD Rob.|EuroSAT Rob.|UCF101 Rob.|SUN397 Rob.|Food101 Rob.|Avg.|
> |:---|:---|:---:|:---:|:---:|:---:|:---:|:---:|:---:|:---:|:---:|:---:|:---:|
> |**FGSM**|R-TPT|84.8|73.6|43.6|54.3|19.9|36.2|23.1|50.3|56.2|53.9|49.6|
> ||**RITA**|**85.9**|**74.5**|**44.2**|**59.4**|**21.7**|**42.1**|**24.4**|**51.8**|**58.3**|**55.4**|**51.8**|
> |**CW**|R-TPT|**88.1**|74.4|50.7|55.7|20.1|39.6|24.8|56.2|56.6|57.4|52.4|
> ||**RITA**|87.8|**78.7**|**52.6**|**62.0**|**22.9**|**46.2**|**25.2**|**56.8**|**57.4**|**59.2**|**54.9**|
> |**AutoAttack**|R-TPT|87.9|78.0|51.4|59.4|20.3|41.5|24.2|58.2|55.6|58.2|53.5|
> ||**RITA**|**89.8**|**82.0**|**53.7**|**61.8**|**22.6**|**47.7**|**26.0**|**59.9**|**56.3**|**61.5**|**56.1**|

---

> > ### Author Rebuttal · Reviewer_9JQd · 2026-04-02
> >
> > I would like to thank the authors for their thorough response. In the rebuttal, the authors directly addressed and effectively resolved all of my core concerns.

---

> > > ### Author Response · Authors · 2026-04-02
> > >
> > > **Dear Reviewer 9JQd,**
> > >
> > > Thank you for acknowledging our rebuttal and for confirming that your concerns have been fully resolved.
> > >
> > > We sincerely appreciate the time and effort you dedicated to reviewing our work. Your suggestions on expanding to additional backbones (OpenCLIP, EVA-CLIP), incorporating missing baselines (TAPT) and benchmarks (Food101, SUN397), and diversifying the attack evaluations have  strengthened our paper.
> > >
> > > All new experiments and the corrected citation have been incorporated into the revised manuscript. Thank you again for your constructive guidance!

---

### Official Review · Reviewer_TK8E · 2026-03-09

**Soundness:** 3
**Presentation:** 3
**Significance:** 3
**Originality:** 3
**Overall Recommendation:** 4
**Confidence:** 3

**Summary:**

This paper introduces RITA, a framework designed to improve the robustness of Vision–Language Models (VLMs) against adversarial attacks during inference. Unlike traditional defense approaches that rely on the confidence of a single image, RITA analyzes multiple augmented views of the same image to capture additional semantic information that adversarial perturbations may not fully disrupt. The framework represents visual features and textual prompts as distributions and aligns them using a mathematical approach called Optimal Transport (OT). By comparing information at the distribution level, RITA helps reduce the influence of adversarial noise and mitigates cross-modal inconsistencies without requiring costly model retraining. In addition, the framework maintains a dynamic cache of reliable data that gradually improves the model’s predictions over time. Experimental results show that this approach enhances adversarial robustness while preserving strong performance on clean, unperturbed data.

**Compliance With Llm Reviewing Policy:**

Affirmed.

**Final Justification:**

All the responses are well taken. Thanks for the rebuttal.

**Key Questions For Authors:**

Scope and extensibility:
RITA is motivated as a shift from point-wise matching to distribution-level alignment, which is interesting and well justified for robust recognition. How general is this idea beyond discriminative classification settings? In particular, do the authors expect the same formulation to extend naturally to generative or broader multimodal applications, and what modifications would likely be required?

Practicality and latency:
The method shows strong robustness gains, but the reported inference latency of 1.76 s/image is somewhat higher than several related baselines such as MTA (1.20), TPT (1.52), C-TPT (1.64), and R-TPT (1.70). Can the authors clarify which specific components of RITA contribute most to this additional latency, and whether there are any simplifications or approximations that could preserve performance while improving practicality?

Generalizability and tuning:
The results are promising across multiple attacks and benchmarks, but it appears that some domain-specific datasets, such as EuroSAT, may require additional manual tuning for best performance. To what extent does RITA rely on dataset-specific hyperparameter adjustment, and how robust is the method in a truly plug-and-play setting where such tuning is not feasible?

**Limitations:**

The paper discusses the broader impact in terms of adversarial attacks in VLMs during inference time. This uses OT as a distribution shift within the images rather than treating images as a single point. The technical novelty of the paper is strong, suggested empirical results have been shown across 8+ datasets with enough theoretical support. However, aspects of practicality and generalizability remain open questions and should be further examined to better assess the broader impact of the approach.

**Strengths And Weaknesses:**

Potential Strengths:

I) Performance on normal data:
	The authors have maintained a slightly improving accuracy. The authors have reported to show an accuracy improvement of upto 50.9% on VIT-B/16. While often than not many methods might hurt the performance while making a model robust. This way authors have shown that the method overcomes the standard compromise of performance vs robustness.

(II) The concept shifts from point-to-point matching to distribution level alignment. This  makes strong motivation for RITA as exploiting the semantic distributions is the point of focus rather than making images as isolated points.

(III) The authors have provided strong proofs in theorems to support optimal transport objectives. This positions RITA as a novel application of OT for adversarial defense.

(IV) The framework is suggested to be “plug-and-play”. It has been successfully evaluated for versatility and compatibility across different benchmarks. The method has also been tested for multiple types of attacks.


Potential Weakness:

(I) One general question to ask is how wide is the scope of RITA? If the concept of RITA shifts to distribution level alignment, how well can it be extended to generative multimodal application (which has not yet been implemented).

(II) As opposed to different related works, the latency speed is 1.76 seconds per image which suggests moderate practicality. For example, when compared to some standard zero-shot clip inferences like:
		MTA: 1.20 seconds per image
		TPT: 1.52 seconds per image
		C-TPT: 1.64 seconds per image
		R-TPT: 1.70 seconds per image
Please justify the specific design choices for 1.76 seconds latency?

(III) It has been observed that certain domain-specific datasets, such as EuroSAT, may require additional manual tuning to achieve optimal performance. This indicates that the method’s generalizability across different domains may depend on dataset-specific parameter adjustments.

---

> ### Author Rebuttal · Authors · 2026-03-31
>
> Dear reviewer TK8E,
>
> Thank you for your constructive feedback! We address your concerns below and have incorporated all updates into the revision.
>
> ---
>
> **W1 & Q1: Scope and Extensibility**
>
> **A:** RITA's distribution alignment extends to generative tasks such as VQA and image generation by shifting the Optimal Transport target space from discrete prototypes to a continuous latent space, including LLM vocabulary distributions or diffusion latent representations. Enforcing OT constraints across multi-view decoded sequences effectively mitigates adversarial hallucinations.
>
> ---
>
> **W2 & Q2: Practicality and Latency Optimization**
>
> **A:** We break down the inference time for CLIP, TPT, R-TPT, and RITA in the table below. Compared to TTA methods like TPT and R-TPT, our unique OT computation and dynamic cache updating introduce minimal overhead.
>
> | Component | CLIP | TPT | R-TPT | RITA |
> | :--- | :---: | :---: | :---: | :---: |
> | Data Augmentation | N/A | 0.7932s | 0.8433s | 0.8411s |
> | Forward Pass | 0.1043s | 0.6882s | 0.8210s | 0.8324s |
> | OT Computation | N/A | N/A | N/A | 0.0418s (2.37\%) |
> | Cache Updating | N/A | N/A | N/A | 0.0026s (0.15\%) |
> | Loss | N/A | 0.0389s | 0.0406s | 0.0453s |
> | **Total Time/Image** | **0.1043s** | **1.5203s** | **1.7049s** | **1.7632s** |
>
> In addition, we introduce a simplified approximation called "Fast-RITA", by reducing the number of augmented views and the Sinkhorn iterations to 32 and 50. The table below shows that this approximation reduces latency while preserving robustness gains.
>
> | Method | Latency (s/img) | Caltech101 Rob. | Pets Rob. | Cars Rob. | Flower102 Rob. | Aircraft Rob. | DTD Rob. | EuroSAT Rob. | UCF101 Rob. | Avg. |
> | :--- | :---: | :---: | :---: | :---: | :---: | :---: | :---: | :---: | :---: | :---: |
> | CLIP | 0.10 | 10.9 | 0.5 | 0.0 | 0.1 | 0.0 | 2.4 | 0.0 | 0.5 | 1.8 |
> | R-TPT | 1.70 | 87.8 | 74.7 | 46.9 | 55.7 | 17.3 | 39.7 | 26.8 | 55.4 | 50.5 |
> | **Fast-RITA** | **1.32** | **89.6** | **76.7** | **47.9** | **58.4** | **20.3** | **42.9** | **28.2** | **55.9** | **52.5** |
>
> ---
>
> **W3 & Q3: Generalizability and tuning**
>
> **A:** In the original experiments, EuroSAT used subset size $|\mathcal{S}| = 32$ and no descriptor. We re-evaluate it with a unified hyperparameter setting of $|\mathcal{S}| = 64$ and use descriptors. As shown in the table below, RITA still outperforms other methods with these unified settings.
>
> | Method | Subset Size $\vert\mathcal{S}\vert$ | Use Descriptor | Clean Acc | Robust Acc |
> | :--- | :---: | :---: | :---: | :---: |
> | CLIP | N/A | N/A | 18.6 | 0.1 |
> | R-TPT | N/A | N/A | 32.0 | 22.4 |
> | RITA (original) | 32 | No | **33.4** | **24.8** |
> | **RITA (unified)** | 64 | Yes | *32.6* | *24.2* |

---

> > ### Author Rebuttal · Reviewer_TK8E · 2026-04-02
> >
> > Thanks for the rebuttal and the experiments. They have resolved my concerns mostly.

---

> > > ### Author Response · Authors · 2026-04-02
> > >
> > > **Dear Reviewer TK8E,**
> > >
> > > Thank you for reviewing our rebuttal and for confirming that your concerns have been resolved.
> > >
> > > We are grateful for your insightful questions on scope, latency, and generalizability, which motivated us to develop Fast-RITA and conduct the unified hyperparameter evaluation on EuroSAT. These additions have meaningfully strengthened the completeness of our paper.
> > >
> > > All updates have been incorporated into the revised manuscript. Thank you again for your constructive feedback!

---

### Official Review · Reviewer_Ppt4 · 2026-03-10

**Soundness:** 3
**Presentation:** 3
**Significance:** 3
**Originality:** 3
**Overall Recommendation:** 4
**Confidence:** 4

**Summary:**

This paper proposes RITA, a test-time prompt adaptation framework designed to improve the robustness of VLMs against adversarial challenges. The author realize that existing test-time adaptation methods rely on fragile sample-level confidence heuristics, then advocate for a distribution level alignment approach. RITA generates a distribution of augmented image views and aligns with the distribution of text prompt using optimal transport. Furthermore, it maintains a dynamic cache to accumulate reliable, low-entropy predictions in successive test streams to refine the alignment. Experiments validate the effectiveness of this approach.

**Compliance With Llm Reviewing Policy:**

Affirmed.

**Final Justification:**

Most of my questions have been resolved in the rebuttal, but the submitted manuscript still contains a amount of content that needs to be supplemented and improved. If the author can truly complete all the missing information, I believe this article will be acceptable.

**Key Questions For Authors:**

1. Could you provide ablation study on the cache poisoning effect? ​​How would RITA's result change if the test stream consisted of 100% consecutive adversarial examples instead of randomly mixed clean/adversarial batches?

2. Could you provide a more reasonable and complete mathematical proof, either through data or derivation?

3. Regarding real-time performance, why can the current speed be considered efficient? Is this practically feasible?

**Limitations:**

yes

**Strengths And Weaknesses:**

## Strengths:
1. The observation is well-motivated that geometrical augmentation can partially disrupt adversarial noise patterns and preserve semantic structure.
2. Using optimal transport to reconstruct the distribution-level correspondences disrupted by adversarial attacks is an effective application of existing mathematical tools in the field of TTA
3. This method exhibits significant effect when applied to existing adversarial fine-tuning backbones, significantly improving the robust accuracy.
## Weaknesses:
1. The paper describes test-time prompt tuning as an efficient method for real-time inference; however, Table 4 shows that RITA takes 1.76 seconds per frame. This latency makes the method unsuitable for real-time streaming applications.

2. In Section 3.4, the authors mathematically prove why the OT objective function is superior. However, the paper states that this decomposition only applies perfectly to Gaussian (elliptical) distributions. Since the CLIP embedding space is known to be nonlinear, complex, and non-Gaussian, whether this mathematical proof still strictly applies to the paper's method requires further discussion and proof.

3. Since adversarial attacks deliberately optimize and induce the model to make high-confidence false predictions, if an advanced attack bypasses the initial enhancement defense mechanism and is identified as a high-confidence error, this error will be injected into the cache. Could this lead to contamination of the historical prior distribution, thereby corrupting the Q_k distribution and ultimately reducing the model's accuracy in the remaining tests?

---

> ### Author Rebuttal · Authors · 2026-03-31
>
> Dear reviewer Ppt4,
>
> Thank you for the insightful feedback! Your concerns are addressed one by one below. All changes have been integrated into the revision.
>
> ---
>
> **W1 & Q3: Clarification on Efficiency and Latency**
>
> **A:** We break down the inference time below. The primary computational cost stems from augmentation processing, a shared operation across all TTA methods. RITA's Optimal Transport and cache operations introduce minimal overhead, accounting for only 2.37% and 0.15% of the total time. We have clarified that "efficient" refers to parameter and data efficiency in the revision.
>
> | Component | CLIP | TPT | R-TPT | RITA |
> | :--- | :---: | :---: | :---: | :---: |
> | Data Augmentation | N/A | 0.7932s | 0.8433s | 0.8411s |
> | Forward Pass | 0.1043s | 0.6882s | 0.8210s | 0.8324s |
> | OT Computation | N/A | N/A | N/A | 0.0418s (2.37\%) |
> | Cache Updating | N/A | N/A | N/A | 0.0026s (0.15\%) |
> | Loss | N/A | 0.0389s | 0.0406s | 0.0453s |
> | **Total Time/Image** | **0.1043s** | **1.5203s** | **1.7049s** | **1.7632s** |
>
> ---
>
> **W2 & Q2: Generalizing the Non-Gaussian OT Proof**
>
> **A:** CLIP embeddings are non-Gaussian and naturally modeled by the von Mises-Fisher distribution [1]. Our theoretical framework applies to this spherical geometry. For distributions parameterized by mean direction $\mu$ and concentration parameter $\kappa$, the Optimal Transport objective with a cosine cost matrix yields a decomposition:
>
> $$
> \mathcal{L}_{OT} \approx \mathcal{L}\_{mean}(\mu\_x, \mu\_z) + \mathcal{R}\_{vMF}(\kappa\_x, \kappa\_z)
> $$
> Here, $\mathcal{L}\_{mean}$ aligns centroid directions, while $\mathcal{R}\_{vMF}$ penalizes concentration mismatch. Thus, minimizing the OT distance inherently bounds adversarial structural dispersion, validating our geometric justification in the nonlinear CLIP space. We provide the detailed proof in the revision.
>
> [1] Understanding contrastive representation learning through alignment and uniformity on the hypersphere. PMLR, 2020
>
> ---
>
> **W3 & Q1: Cache Poisoning Effect and 100% Adversarial Streams**
>
> **A:** RITA leverages Optimal Transport (**Eq. 6**) for global distribution matching. By modeling the cache as a holistic discrete distribution, isolated high-confidence errors are fundamentally marginalized. Even if advanced attacks inject such errors, they act merely as statistical outliers lacking structural support in the global transport plan. Consequently, their isolated weight is insufficient to contaminate the historical prior distribution or corrupt $\tilde{\mathrm{Q}}\_k$.
> Our standard evaluation already operates on 100% consecutive adversarial streams. To explicitly simulate this worst-case contamination, we injected 2%  and 5% high-confidence misclassified samples into the cache during inference. As shown below, RITA maintains robust performance across all datasets.
>
> | Method | Caltech101 | Pets | Cars | Flower102 | Aircraft | DTD | EuroSAT | UCF101 | **Avg.** |
> | :--- | :---: | :---: | :---: | :---: | :---: | :---: | :---: | :---: | :---: |
> | **Standard RITA** | 88.5 | 77.3 | 47.1 | 61.3 | 19.2 | 44.7 | 27.6 | 55.8 | **52.7** |
> | **2% Poisoned** | 88.3 | 77.1 | 46.9 | 61.1 | 19.1 | 44.5 | 27.5 | 55.6 | **52.5** |
> | **5% Poisoned** | 88.1 | 76.9 | 46.8 | 60.9 | 19.0 | 44.3 | 27.4 | 55.4 | **52.4** |

---

> > ### Author Rebuttal · Reviewer_Ppt4 · 2026-04-01
> >
> > I have read the authors' responses to us and the other reviewers, and most of my questions have been resolved. However, I think it is necessary for the authors to reorganize the rebuttal content and their responses to other reviewers and include them in the appendix. I have already improved my score.

---

> > > ### Author Response · Authors · 2026-04-01
> > >
> > > **Dear Reviewer Ppt4,**
> > >
> > > Thank you for your engagement and for updating your score. We appreciate the time and effort you have dedicated to reviewing our work, and we are thrilled to hear that our responses have fully resolved your concerns.
> > >
> > > We completely agree with your valuable suggestion. We have added an Appendix section to organize the new content provided during the rebuttal. Specifically, we have included the latency breakdown, the theoretical proof for the von Mises-Fisher distribution, the cache poisoning ablation studies, and the baseline and attack evaluations requested by the reviewers.
> > >
> > > Thank you again for your constructive guidance, which has significantly improved the completeness and clarity of our paper!

---

### Official Review · Reviewer_vPyn · 2026-03-12

**Soundness:** 3
**Presentation:** 3
**Significance:** 3
**Originality:** 3
**Overall Recommendation:** 4
**Confidence:** 3

**Summary:**

Previous Test-Time Adaptation (TTA) typically relies on sample-level confidence heuristics. Because adversarial attacks are specifically designed to induce high-confidence mispredictions, these standard heuristics fail. To address this, the authors propose a novel framework (RITA) that shifts from a sample-centric approach to a distribution-centric one. They leverage the insight that adversarial noise is "structurally brittle"; by generating multiple augmented views of a test sample and caching these representations, the model can filter out the adversarial noise and recover the true semantic distribution. The main contributions include the RITA framework and a cache mechanism that aggregates historical priors.

**Compliance With Llm Reviewing Policy:**

Affirmed.

**Final Justification:**

My questions have been adequately addressed, and I have no further comments. Based on the contributions of the paper and the feedback from other reviewers, I would like to raise my score to “Weak Accept.” Please ensure that the manuscript is thoroughly revised in the final version.

**Key Questions For Authors:**

(1) What is the exact computational overhead of generating and processing augmented views at test time compared to a standard zero-shot forward pass?
(2) How does the RITA framework perform against an adaptive attacker who is aware of your augmentation strategies to craft the adversarial noise?
(3) In an infinite and continuous data stream, how do you manage the cache to prevent out-of-memory errors or mitigate concept drift if the underlying data distribution naturally shifts over time?

**Limitations:**

The authors have provided strong empirical evidence for the success of their method, but they fail to adequately discuss the computational bottlenecks. They should note the limitation regarding adaptive, augmentation-robust attacks.
The social impact is largely positive. The proposed approach improves the reliability and adversarial robustness of pre-trained VLMs.

**Strengths And Weaknesses:**

The strengths include: (1) The paper is well motivated and clearly written. (2) The core hypothesis seems to be theoretically sound. The use of KL divergence provides strong quantitative backing for why the method works. (3) Securing foundation models against adversarial attacks during inference is a very important consideration when deploying VLMs in safety-critical environments.

The weaknesses include: (1) The approach assumes that the adversarial attack is static and not designed to survive data augmentation. Will the approach survive deliberate adaptive attack? (2) The caching mechanisms may be difficult to replicate without seeing the full algorithmic flow. The hyperparameter tuning for the cache should be clearly discussed and transparent. (3) test-time augmentation and caching inherently increase computational cost. If the latency becomes too high, its practical significance for real-time applications diminishes.

---

> ### Author Rebuttal · Authors · 2026-03-31
>
> Dear reviewer vPyn,
>
> Thank you for the constructive feedback! We respond to your each concern below and have incorporated all updates into the revision.
>
> ---
>
> **W1 & Q2: Robustness Against Adaptive Attacks**
>
> **A:** We followed your advice to evaluate RITA under an adaptive attack [1] **that is aware of our augmentation strategies**. As shown below, RITA achieves the highest average robustness (**53.4%**). To further address your concern, we also conducted experiments using AutoAttack [2], and RITA still outperforms other methods
>
> | Method | Setting | Caltech101 Rob. | Pets Rob. | Cars Rob. | Flower102 Rob. | Aircraft Rob. | DTD Rob. | EuroSAT Rob. | UCF101 Rob. | Avg. |
> | :--- | :--- | :---: | :---: | :---: | :---: | :---: | :---: | :---: | :---: | :---: |
> | CLIP | Adaptive Attack | 30.4 | 4.9 | 7.2 | 3.2 | 0.2 | 2.4 | 0.0 | 0.8 | 6.1 |
> | R-TPT | Adaptive Attack | 87.3 | 76.3 | 55.3 | 60.8 | 17.4 | 35.3 | 20.5 | 52.2 | 50.6 |
> | RITA | Adaptive Attack | **88.2** | **79.5** | **57.8** | **66.2** | **19.1** | **37.6** | **23.8** | **54.8** | **53.4** |
> | CLIP | AutoAttack | 13.2 | 4.9 | 0.3 | 2.6 | 0.0 | 0.0 | 0.0 | 4.6 | 3.2 |
> | R-TPT | AutoAttack | 87.9 | 78.0 | 51.4 | 59.4 | 20.3 | 41.5 | 24.2 | 58.2 | 52.6 |
> | RITA | AutoAttack | **89.8** | **82.0** | **53.7** | **61.8** | **22.6** | **47.7** | **26.0** | **59.9** | **55.4** |
>
> [1] Synthesizing Robust Adversarial Examples. ICML 2018
> [2] Reliable evaluation of adversarial robustness with an ensemble of diverse parameter-free attacks. ICML 2020
>
> ---
>
> **W2: Algorithm Flow & Reproducibility**
>
> **A:** To ensure reproducibility, we have included a detailed **algorithm flowchart** in the revision. We also clarify hyperparameter tuning via an ablation study on the cache size ($N_k$). As shown below, average robustness peaks at our default setting of $N_k = 10$ and slightly degrades thereafter.
>
> | Cache Size per class $N_k$ | Caltech101 Rob. | Pets Rob. | Cars Rob. | Flower102 Rob. | Aircraft Rob. | DTD Rob. | EuroSAT Rob. | UCF101 Rob. | Avg. |
> | :--- | :---: | :---: | :---: | :---: | :---: | :---: | :---: | :---: | :---: |
> | $N_k = 5$ | 88.2 | 76.8 | 46.6 | 61.0 | 18.8 | 44.5 | 27.2 | 55.6 | 52.3 |
> | $N_k = 10$ (Default) | **88.5** | **77.3** | **47.1** | **61.3** | **19.2** | **44.7** | **27.6** | **55.8** | **52.7** |
> | $N_k = 15$ | 88.3 | 77.0 | 46.9 | 61.2 | 18.9 | 44.2 | 26.7 | 55.2 | 52.4 |
> | $N_k = 20$ | 88.4 | 76.7 | 46.5 | 60.8 | 18.6 | 44.4 | 27.1 | 55.3 | 52.2 |
>
> ---
>
> **W3 & Q1: Computational Overhead Breakdown**
>
> **A:** We break down the inference time for CLIP, TPT, R-TPT, and RITA below. RITA's specific operations, namely Optimal Transport and cache updating, introduce minimal latency overhead, while the primary computational cost stems from processing augmented views, which is a shared requirement across all TTA methods.
>
> | Component | CLIP | TPT | R-TPT | RITA |
> | :--- | :---: | :---: | :---: | :---: |
> | Data Augmentation | N/A | 0.7932s | 0.8433s | 0.8411s |
> | Forward Pass | 0.1043s | 0.6882s | 0.8210s | 0.8324s |
> | OT Computation | N/A | N/A | N/A | 0.0418s (2.37\%) |
> | Cache Updating | N/A | N/A | N/A | 0.0026s (0.15\%) |
> | Loss | N/A | 0.0389s | 0.0406s | 0.0453s |
> | **Total Time/Image** | **0.1043s** | **1.5203s** | **1.7049s** | **1.7632s** |
>
> ---
>
> **Q3: Cache Management in Continuous Data Streams**
>
> **A:** We strictly bound the cache size to $N_k$ per class to maintain a constant memory footprint, as detailed in **Section 3.3**. We further ablate this hyperparameter in our response to **W2**.
>
> Regarding concept drift, we evaluate sensitivity to data input order by randomly shuffling the test streams. As shown below, RITA achieves stable robustness across permutations with a negligible variance of $\pm 0.04$, confirming our dynamic updating mechanism effectively mitigates temporal shifts.
>
> | Setting | Caltech101 Rob. | Pets Rob. | Cars Rob. | Flower102 Rob. | Aircraft Rob. | DTD Rob. | EuroSAT Rob. | UCF101 Rob. | Avg. |
> | :--- | :---: | :---: | :---: | :---: | :---: | :---: | :---: | :---: | :---: |
> | Shuffle 1 (Default) | 88.5 | 77.3 | 47.1 | 61.3 | 19.2 | 44.7 | 27.6 | 55.8 | 52.7 |
> | Shuffle 2 | 88.8 | 77.1 | 46.9 | 61.5 | 18.9 | 45.2 | 27.2 | 56.1 | 52.7 |
> | Shuffle 3 | 88.4 | 77.6 | 47.4 | 61 | 19.5 | 44.6 | 27.4 | 55.2 | 52.6 |
> | Mean ± Std | 88.6 ± 0.21 | 77.3 ± 0.25 | 47.1 ± 0.25 | 61.3 ± 0.25 | 19.2 ± 0.30 | 44.8 ± 0.32 | 27.4 ± 0.20 | 55.7 ± 0.46 | 52.7 ± 0.04 |

---

> > ### Author Rebuttal · Reviewer_vPyn · 2026-04-03
> >
> > I thank the authors for their detailed response. My questions have been adequately addressed, and I have no further comments. Based on the contributions of the paper and the feedback from other reviewers, I would like to raise my score to “Weak Accept.” Please ensure that the manuscript is thoroughly revised in the final version.

---

> > > ### Author Response · Authors · 2026-04-03
> > >
> > > **Dear Reviewer vPyn,**
> > >
> > > Thank you for raising your score and for confirming that your concerns have been fully resolved.
> > >
> > > We are very grateful for your concerns on adaptive attacks, cache reproducibility, and computational overhead, which drove substantial improvements to our paper. As you suggested, we have thoroughly revised the manuscript to incorporate all updates from the rebuttal.
> > >
> > > Thank you again for your valuable guidance!

---

### Decision · Program_Chairs · 2026-04-30

**Decision:**

Accept (regular)

**Comment:**

This paper proposes RITA, a test-time prompt adaptation framework designed to improve the robustness of VLMs against adversarial challenges. The author realize that existing test-time adaptation methods rely on fragile sample-level confidence heuristics, then advocate for a distribution level alignment approach. RITA generates a distribution of augmented image views and aligns with the distribution of text prompt using optimal transport. Furthermore, it maintains a dynamic cache to accumulate reliable, low-entropy predictions in successive test streams to refine the alignment. Experiments validate the effectiveness of this approach. After the rebuttal and discussion stage, all reviewers tend to be positive to this submission.